# On the Last-iterate Convergence in Time-varying Zero-sum Games: Extra Gradient Succeeds where Optimism Fails

**Yi Feng**
SUFE*
2021310186@live.sufe.edu.cn

**Hu Fu**
SUFE[†]
fuhu@shufe.edu.cn

**Qun Hu**
SUFE
huqun29@163.com

**Ping Li**
SUFE
lping0423@163.com

**Ioannis Panageas**
University of California, Irvine
ipanagea@ics.uci.edu

**Bo Peng**
SUFE
ahqspb@163.sufe.edu.cn

**Xiao Wang**
SUFE[†]
wangxiao@shufe.edu.cn

## Abstract

Last-iterate convergence has received extensive study in two player zero-sum games starting from bilinear, convex-concave up to settings that satisfy the MVI condition. Typical methods that exhibit last-iterate convergence for the aforementioned games include extra-gradient (EG) and optimistic gradient descent ascent (OGDA). However, all the established last-iterate convergence results hold for the restrictive setting where the underlying repeated game does not change over time. Recently, a line of research has focused on regret analysis of OGDA in time-varying games, i.e., games where payoffs evolve with time; the last-iterate behavior of OGDA and EG in time-varying environments remains unclear though. In this paper, we study the last-iterate behavior of various algorithms in two types of unconstrained, time-varying, bilinear zero-sum games: periodic and convergent perturbed games. These models expand upon the usual repeated game formulation and incorporate external environmental factors, such as the seasonal effects on species competition and vanishing external noise. In periodic games, we prove that EG will converge while OGDA and momentum method will diverge. This is quite surprising, as to the best of our knowledge, it is the first result that indicates EG and OGDA have qualitatively different last-iterate behaviors and do not exhibit similar behavior. In convergent perturbed games, we prove all these algorithms converge as long as the game itself stabilizes with a faster rate than $1/t$.

## 1 Introduction

A central problem in game theory and min-max optimization is to come up with a pair of vectors $(x, y)$ that solves

$$\min_{x \in \mathcal{X}} \max_{y \in \mathcal{Y}} x^\top A y, \tag{1}$$

---

Authors are listed according to the alphabetical order.

*Shanghai University of Finance and Economics
[†]Key Laboratory of Interdisciplinary Research of Computation and Economics, Ministry of Education.

37th Conference on Neural Information Processing Systems (NeurIPS 2023).

where $\mathcal{X} \subset \mathbb{R}^n$ and $\mathcal{Y} \subset \mathbb{R}^m$ are convex sets, and $A$ is a $n \times m$ payoff matrix. The above captures two-player zero-sum games in which $x^\top A y$ is interpreted as the payment of the "min player" $x$ to the "max player" $y$. If $\mathcal{X} = \mathbb{R}^n$ and $\mathcal{Y} = \mathbb{R}^m$ the setting is called *unconstrained*, otherwise it is *constrained*. Soon after the minimax theorem of Von Neumann was established (for compact $\mathcal{X}, \mathcal{Y}$), learning dynamics such as fictitious play (Brown [1951]) were proposed for solving min-max optimization problems. Blackwell's approachability theorem ( Blackwell [1956]) further propelled the field of online learning, which lead to the discovery of several learning algorithms; such learning methods include multiplicative-weights-update method, online gradient descent/ascent and their optimistic variants and extra-gradient methods.

**Last Iterate Convergence.**   There have been a vast literature on whether or not the aforementioned dynamics converge in an average sense or exhibit last-iterate convergence when applied to zero-sum games. Dating back to Nesterov (Nesterov [2005]), there have been quite a few results showing that online learning algorithms have last-iterate convergence to Nash equilibria in zero-sum games. Examples include optimistic multiplicative weights update (Daskalakis and Panageas [2019], Wei et al. [2021]), optimistic gradient descent ascent (OGDA) (Daskalakis et al. [2018], Liang and Stokes [2019a]) (applied even to GANs) for unconstrained zero-sum games, OGDA for constrained zero-sum games (Wei et al. [2021], Cai et al. [2022], Gorbunov et al. [2022b]) and extra-gradient methods (Mertikopoulos et al. [2019], Cai et al. [2022], Gorbunov et al. [2022a]) using various techniques, including sum of squares.

Nevertheless, all aforementioned results assume that the underlying repeated zero-sum game remains invariant throughout the learning process. In many learning environments that assumption is unrealistic, see (Duvocelle et al. [2022], Mai et al. [2018], Cardoso et al. [2019]) and references therein. One more realistic learning setting is where the underlying game is actually changing; this game is called time-varying. There have been quite a few works that deal with time-varying games, where they aim at analyzing the duality gap or dynamic regret (Zhang et al. [2022]) and references therein for OGDA and variants. However, in all these prior works, last-iterate convergence has not been investigated; the main purpose of this paper is to fill in this gap. We aim at addressing the following question:

*Will learning algorithms such as optimistic gradient descent ascent or extra-gradient exhibit last-iterate converge in time-varying zero-sum games?*

**Our contributions.**   We consider unconstrained two-player zero-sum games with a time-varying payoff matrix (that is the payoff matrix $A_t$ depends on time $t$)

$$\min_{x \in \mathbb{R}^n} \max_{y \in \mathbb{R}^m} x^\top A_t y, \qquad \text{(Time-varying zero sum game)}$$

in which the payoff matrix $A_t$ varies with time in the following two ways:

- **Periodic games:** $A_t$ is a periodic function with period $T$, i.e., $A_{t+T} = A_t$.

- **Convergent perturbed games:** $A_t = A + B_t$, $\lim_{t \to \infty} B_t = 0$.

In a repeated time-varying zero-sum game, players choose their learning algorithms and repeatedly play the zero-sum game. In the $t$-th round, when the players use strategy $(x_t, y_t)$, they receive their payoff $-A_t^\top x_t$ and $A_t y_t$.

In this paper we show the following results:

- For **periodic games:** We prove that when two players use extra-gradient, their strategies will converge to the common Nash equilibrium of the games within a period with an exponential rate, see Theorem 3.1 for details. Additionally, we provide an example where optimistic gradient descent ascent and negative momentum method diverge from the equilibrium with an exponential rate, see Theorem 3.2.

  To the best of our knowledge, this is the first result that provides a clear separation between the behavior of extra-gradient methods and optimistic gradient descent ascent.

- For **convergent perturbed games:** Assuming $\sum_{t=1}^{\infty} \|B_t\|_2$ is bounded, we prove that the extra-gradient, optimistic gradient descent ascent, and negative momentum method all converge to the Nash equilibrium of the game defined by payoff matrix $A$ with a rate determined by $\{B_t\}_t$ and singular values of $A$, see Theorem 3.3. Furthermore, we prove that extra-gradient will asymptotically converge to equilibrium without any additional assumptions on perturbations besides $\lim_{t \to \infty} B_t = 0$, see Theorem 3.4.

**Related work on time-varying games.** The closest work to ours that argues about stabilization of mirror descent type dynamics on convergent strongly monotone games is (Duvocelle et al. [2022]). Most of the literature has been focused on proving either recurrence/oscillating behavior of learning dynamics in time-varying periodic games (Mai et al. [2018], Fiez et al. [2021] ) and references therein or performing regret analysis (Roy et al. [2019], Cardoso et al. [2019], Zhang et al. [2022]). In particular, the latter work extends results on RVU (Syrgkanis et al. [2015]) bounds to argue about dynamic regret (Zinkevich [2003]).

**Technical Comparison.** We investigate the last iterate behaviors of learning dynamics through their formulation of linear difference systems. This approach has also been used to establish last iterate convergence results for learning dynamics in time-independent games (Zhang and Yu [2020], Liang and Stokes [2019b], Gidel et al. [2019]). One common key point of these works is to prove that a certain matrix has no eigenvalues with modulus larger than 1, then the last iterate convergence can be guaranteed by the general results of autonomous difference systems. However, this method cannot be generalized to the time-varying games where the corresponding difference systems are non-autonomous. In particular, the dynamical behavior of a non-autonomous system is not determined by the eigenvalues of a single matrix. In fact, it is difficult to establish convergence/divergence results even for non-autonomous system with special structures, such as periodic or perturbed systems. In this paper, to get such results, we employ both general results in linear difference systems, such as Floquet theorem and Gronwall inequality, and special structures of the difference systems associated to learning dynamics.

**Organization.** In Section 2, we present the necessary background for this work. The main results are stated in Section 3. In Section 4, we present numerical experiments and in Section 5, we conclude with a discussion and propose some future research problems.

# 2 Preliminaries

## 2.1 Definitions

**Zero-sum blinear game.** An unconstrained two players zero-sum game consists of two agents $\mathcal{N} = \{1, 2\}$, and losses of both players are determined via payoff matrix $A \in \mathbb{R}^{n \times m}$. Given that player 1 selects strategy $x \in \mathbb{R}^n$ and player 2 selects strategy $y \in \mathbb{R}^m$, player 1 receives loss $u_1(x, y) = \langle x, Ay \rangle$, and player 2 receives loss $u_2(x, y) = -\langle y, A^\top x \rangle$. Naturally, players want to minimize their loss resulting the following min-max problem:

$$\min_{x \in \mathbb{R}^n} \max_{y \in \mathbb{R}^m} x^\top A y \qquad \text{(Zero-Sum Game)}$$

Note that the set $\{(x^*, y^*) | A^\top x^* = 0, Ay^* = 0\}$ represents the set of equilibrium of the game.

**Time-varying zero-sum bilinear game.** In this paper, we study games in which the payoff matrices vary over time and we define two kinds of such time-varying games.

**Definition 2.1** (Periodic games)**.** *A periodic game with period $T$ is an infinite sequence of zero-sum bilinear games $\{A_t\}_{t=0}^{\infty} \subset \mathbb{R}^{n \times m}$, and $A_{t+T} = A_t$ for all $t \geq 0$.*

Note that the periodic game defined here is the same as Definition 1 in (Fiez et al. [2021]) except for the fact that we are considering a discrete time setting. Therefore, we do not make the assumption that payoff entries are smoothly dependent on $t$.

**Definition 2.2** (Convergent perturbed games)**.** *A convergent perturbed game is an infinite sequence of zero-sum bilinear games $\{A_t\}_{t=0}^{\infty} \subset \mathbb{R}^{n \times m}$, and $\lim_{t \to \infty} A_t = A$ for some $A \in \mathbb{R}^{n \times m}$. Equivalently, write $A_t = A + B_t$, then $\lim_{t \to \infty} B_t = 0$. We will refer to the zero-sum bilinear game defined by $A$ as **stable game**.*

**Learning dynamics in games.** In this paper, we consider three kinds of learning dynamics : optimistic gradient descent ascent (OGDA), extra-gradient (EG) , and negative momentum method. All these methods possess the last-iterate convergence property in repeated game with a time-independent payoff matrix, as demonstrated in previous literature. However, here we state their forms within a time-varying context.

**Optimistic gradient descent-ascent.** We study the optimistic descent ascent method (OGDA) defined as follows:

$$x_{t+1} = x_t - 2\eta A_t y_t + \eta A_{t-1} y_{t-1}, \tag{OGDA}$$
$$y_{t+1} = y_t + 2\eta A_t^\top x_t - \eta A_{t-1}^\top x_{t-1}.$$

Optimistic gradient descent ascent method was proposed in (Popov [1980]), and here we choose the same parameters as (Daskalakis et al. [2018]). The last iterate convergence property of OGDA in unconstrained bilinear game with a time-independent payoff was proved in (Daskalakis et al. [2018]). Recently, there are also works analyzing the regret behaviors of OGDA under a time varying setting (Zhang et al. [2022], Anagnostides et al. [2023]).

**Extra gradient.** We study the extra gradient descent ascent method (EG) defined as follows:

$$x_{t+\frac{1}{2}} = x_t - \gamma A_t y_t, \;\; y_{t+\frac{1}{2}} = y_t + \gamma A_t^\top x_t, \tag{EG}$$
$$x_{t+1} = x_t - \alpha A_t y_{t+\frac{1}{2}}, \;\; y_{t+1} = y_t + \alpha A_t^\top x_{t+\frac{1}{2}}.$$

Note that the extra-gradient method first calculates an intermediate state before proceeding to the next state. Extra-gradient was firstly proposed in (Korpelevich [1976]) with the restriction that $\alpha = \gamma$. Here we choose the parameters same as in (Liang and Stokes [2019a]), where the linear convergence rate of extra-gradient in the bilinear zero-sum game with time-independent was also proven. Convergence of extra-gradient on convex-concave game was analyzed in (Nemirovski [2004], Monteiro and Svaiter [2010]), and convergence guarantees for special non-convex-non-concave game was provided in (Mertikopoulos et al. [2019]).

**Negative momentum method.** We study the alternating negative momentum method (NM), defined as follows:

$$x_{t+1} = x_t - \eta A_t y_t + \beta_1(x_t - x_{t-1}), \tag{NM}$$
$$y_{t+1} = y_t + \eta A_{t+1}^\top x_{t+1} + \beta_2(y_t - y_{t-1}),$$

where $\beta_1, \beta_2 \leq 0$ are the momentum parameters.

Applications of negative momentum method in game optimization was firstly proposed in (Gidel et al. [2019]). Note that the algorithm has an alternating implementation: the update rule of $y_{t+1}$ uses the payoff $A_{t+1}^\top x_{t+1}$, thus in each round, the second player chooses his strategy after the first player has chosen his. It was shown in (Gidel et al. [2019]) that both the negative momentum parameters and alternating implementations are crucial for convergence in bilinear zero-sum games with time-independent payoff matrices. Analysis of the convergence rate of negative momentum method in strongly-convex strongly-concave games was provided in (Zhang and Wang [2021]).

## 2.2 Difference systems

The analysis of the last-iterate behavior of learning algorithms can be reduced to analyzing the dynamical behavior of the associated linear difference systems (Zhang and Yu [2020], Daskalakis and Panageas [2018]). When the payoff matrix is time-independent, the associated difference systems is autonomous. However, as we are studying games with time-varying payoff matrices, we have to deal with non-autonomous difference systems. In general, the convergence behavior of non-autonomous difference systems is much more complicated than that of autonomous ones (Colonius and Kliemann [2014]).

**Linear difference system.** Given a sequence of iterate matrices $\{\mathcal{A}_t\}_{t=1}^\infty \subset \mathbb{R}^{n \times n}$ and initial condition $X_0 \in \mathbb{R}^n$, a linear difference system has form

$$X_{t+1} = \mathcal{A}_t X_t. \tag{Linear difference system}$$

If $\mathcal{A}_t \equiv \mathcal{A}$ is a matrix independent of time, the system is called an **autonomous system**, otherwise, it is called a **non-autonomous system**. We care about the asymptotic behavior of $X_t$, that is, what can we say about $X_t$ as $t \to \infty$.

**Definition 2.3.** *A point $X$ is called **asymptotically stable** under the linear difference system if*

$$\exists\, \delta > 0, \forall\, Y,\ s.t.\ \|Y - X\|_2 \leq \delta \Rightarrow \lim_{t \to \infty} \|(\prod_{t=1}^{\infty} \mathcal{A}_t)Y - X\|_2 = 0.$$

*Moreover, $X$ is called **exponentially asymptotically stable** if the above limit has an exponentially convergence rate, i.e., $\exists\, \alpha \in (0,1)$, such that*

$$\|(\prod_{t=1}^{s} \mathcal{A}_t)Y - X\|_2 \leq \alpha^s \|Y - X\|_2.$$

It is well known that for autonomous linear systems, being asymptotically stable is equivalent to being exponentially asymptotically stable, as shown in Thm 1.5.11 in (Colonius and Kliemann [2014]). However, this equivalence does not hold for non-autonomous systems. [†]

Linear difference system has a formal solution $X_{T+1} = \prod_{t=0}^{T} \mathcal{A}_t X_0$. However, such a representation does not yield much information about the asymptotic behavior of solution as $t \to \infty$, except in the case of autonomous system. In this paper, we mainly consider two classes of non-autonomous linear difference systems: if the iterate matrix $\mathcal{A}_t$ is a periodic function of $t$, the system is called a periodic system; and if $\mathcal{A}_t$ has a limit as $t$ tends to infinity, the system is called a perturbed system.

**Periodic linear system.** If the iterate matrix $\mathcal{A}_t$ in a linear difference system satisfies $\mathcal{A}_t = \mathcal{A}_{t+\mathcal{T}}$, $\forall t \in \mathbb{Z}$, then the system is called a periodic linear system. Denote $\widetilde{\mathcal{A}} = \prod_{j=1}^{\mathcal{T}} \mathcal{A}_{\mathcal{T}-j}$. For a $T$-periodic equation, the eigenvalues $\alpha \in \mathbb{C}$ of $\widetilde{\mathcal{A}}$ is called the Floquet multipliers, and the Floquet exponents are defined by $\lambda_j = \frac{1}{\mathcal{T}} \ln(|\alpha_j|)$. The following Floquet theorem characterizes the stability of a periodic linear difference equation.

**Proposition 2.4** (Floquet theorem, (Colonius and Kliemann [2014]))**.** *The zero solution of a periodic linear difference equation is asymptotically stable if and only if all Floquet exponents are negative.*

Although Floquet theorem provides a method to determine whether a periodic system converges in general, considerable further work may be necessary in order to obtain explicit convergence criteria for specific equation. That is because even if we know the modulus of the largest eigenvalue of each $\mathcal{A}_t$, it is usually difficult to compute the the modulus of the largest eigenvalue of each $\prod_{t=0}^{\mathcal{T}} \mathcal{A}_t$ due to the complex behavior of eigenvalues under matrix multiplication.

**Perturbed linear system.** If the iterative matrix $\mathcal{A}_t$ in a linear difference system satisfies $\mathcal{A}_t = \mathcal{A} + \mathcal{B}_t$ and $\lim_{t \to \infty} \mathcal{B}_t = 0$, then the system is called a perturbed linear system. The convergence behavior of a perturbed linear system is not clear in the literature. A general result in this direction is the following Perron's theorem:

**Theorem 2.5** (Perron's theorem, (Pituk [2002]))**.** *If $X_n$ is a solution of a perturbed linear system, then either $X_n = 0$ for all sufficient large $n$ or $\rho = \lim_{n \to \infty} \sqrt[n]{\|X_n\|_2}$ exists and is equal to the modulus of one of the eigenvalues of matrix $\mathcal{A}$.*

This result can only guarantee the convergence of $X_n$ when all eigenvalues of $\mathcal{A}$ have modulus smaller than 1. In this case, $\lim_{n \to \infty} X_n = 0$. However, for analyzing the non-autonomous linear systems associated to learning dynamics, it is not sufficient as we will show that the stablized matrix of these systems generally has eigenvalues equal to 1.

# 3 Main results

In this section, we present our main results. We present the relationship between learning dynamics and linear difference equation in Section 3.1, investigate the last-iterate behaviors of learning dynamics in a periodic game in Section 3.2, and investigate the last-iterate behaviors of learning dynamics in a convergent perturbed game in Section 3.3. Proofs are deferred to the appendix.

---

[†]To gain intuition of the differences between autonomous and non-autonomous system, consider the following simple 1-dimension example: $x_{t+1} = (1 - \frac{1}{t+1})x_t$. 0 is a stationary point of this system, but every initial points converges to 0 with a rate $\mathcal{O}(\frac{1}{t})$, thus 0 is asymptotically stable but not exponentially asymptotically stable .

## 3.1 Learning dynamics as linear difference systems

Formalizing learning dynamics as linear difference systems is useful for studying their dynamical behaviors. In the following, we present the formulation of optimistic gradient descent ascent, extra-gradient, and negative momentum method as linear difference systems.

**Proposition 3.1.** *Optimistic gradient descent ascent can be written as the following linear difference system:*

$$
\begin{bmatrix} x_{t+1} \\ y_{t+1} \\ x_t \\ y_t \end{bmatrix} = \begin{bmatrix} I & -2\eta A_t & 0 & \eta A_{t-1} \\ 2\eta A_t^\top & I & -\eta A_{t-1}^\top & 0 \\ I & 0 & 0 & 0 \\ 0 & I & 0 & 0 \end{bmatrix} \begin{bmatrix} x_t \\ y_t \\ x_{t-1} \\ y_{t-1} \end{bmatrix}. \tag{2}
$$

*Extra-gradient can be written as the following linear difference system :*

$$
\begin{bmatrix} x_{t+1} \\ y_{t+1} \end{bmatrix} = \begin{bmatrix} I - \alpha\gamma A_t A_t^\top & -\alpha A_t \\ \alpha A_t^\top & I - \gamma\alpha A_t^\top A_t \end{bmatrix} \begin{bmatrix} x_t \\ y_t \end{bmatrix}. \tag{3}
$$

*Negative momentum method can be written as the following linear difference system:*

$$
\begin{bmatrix} x_{t+1} \\ y_{t+1} \\ x_t \\ y_t \end{bmatrix} = \begin{bmatrix} (1+\beta_1)I & -\eta A_t & -\beta_1 I & 0 \\ \eta(1+\beta_1)A_{t+1}^\top & (1+\beta_2)I - \eta^2 A_{t+1}^\top A_t & -\eta\beta_1 A_{t+1}^\top & -\beta_2 I \\ I & 0 & 0 & 0 \\ 0 & I & 0 & 0 \end{bmatrix} \begin{bmatrix} x_t \\ y_t \\ x_{t-1} \\ y_{t-1} \end{bmatrix}. \tag{4}
$$

It is easy to verify that these linear difference systems are equivalent to their corresponding learning dynamics by directly writing down the matrix-vector product.

We will also refer to the iterative matrix of these linear difference systems as the **iterative matrix of their corresponding learning dynamics**. In the following, we will study the convergence/divergence behaviors of these linear difference systems under the condition that $\{A_t\}_t$ is a periodic game or convergent perturbed game. Note that although an intermediate step $(x_{t+\frac{1}{2}}, y_{t+\frac{1}{2}})$ is required in extra-gradient, it is eliminated in (3).

## 3.2 Periodic games

Recall that in a periodic game with period $\mathcal{T}$, the payoff matrices $\{A_s\}_{s=1}^{\infty}$ satisfy $A_{s+\mathcal{T}} = A_s$ for any $s > 0$. Define

$$
\Delta_{i,t} = \|A_i^\top x_t\|_2 + \|A_i y_t\|_2 \tag{5}
$$

for $i \in [\mathcal{T}]$. As in (Daskalakis and Panageas [2018]), we use $\Delta_{i,t}$ as a measurement of the distance between the current strategy $(x_t, y_t)$ and a Nash equilibrium of the zero-sum bilinear game defined by the kernel space of payoff matrix $A_i$. Note that if $(x^*, y^*)$ is a Nash equilibrium of the game defined by $A_i$, then $(A_i^\top x^*, A_i y^*) = (0, 0)$, and

$$
\Delta_{i,t} = \|A_i^\top (x_t - x^*)\|_2 + \|A_i(y_t - y^*)\|_2
$$

thus when strategy is close to equilibrium, $\Delta_{i,t}$ will be small. Moreover, $\Delta_{i,t} = 0$ if and only if $(x_t, y_t)$ is an equilibrium. In this section, we will consider the convergence/growth rate of $\Delta_{i,t}$ at a function of $t$.

We firstly consider Extra-gradient method. Denote the iterative matrix in the linear difference form of Extra-gradient (3) as $\mathcal{A}_t$. In the following theorem, we prove that if two players use the Extra-gradient method, their strategies will converge to the common Nash equilibrium of games in the period with an exponential rate.

**Theorem 3.1.** *When two players use extra-gradient in a periodic games with period $T$, with step size $\alpha = \gamma < \frac{1}{\sigma}$ where $\sigma = \max\{\sigma' | \sigma'$ is a singular value of $A_i$ for some $i \in [\mathcal{T}]\}$. Then*

$$
\Delta_{i,t} \in \mathcal{O}\left((\lambda_*)^{t/\mathcal{T}} \cdot \mathrm{Poly}(t)\right), \ \forall i \in [\mathcal{T}]
$$

*where $\lambda_* = \max\{ |\lambda| \ | \ \lambda$ is an eigenvalue of $\left(\prod_{t=1}^{\mathcal{T}} \mathcal{A}_t\right), \lambda \neq 1\}$, and $\lambda_* < 1$.*

Note that in a periodic game, the iterative matrices of learning dynamics are also periodic, which means that the learning difference systems of these learning dynamics are periodic systems. According to Floquet theorem, see proposition 2.4, the study of dynamical behaviors of a periodic system can be reduced to the autonomous system whose asymptotic behavior is determined by $\prod_{t=1}^{\mathcal{T}} \mathcal{A}_t$.

The key point on the proof of Theorem 3.1 is an observation that the iterative matrix of extra-gradient is a normal matrix, which makes it possible to calculate the Jordan normal form of $\prod_{t=1}^{\mathcal{T}} \mathcal{A}_t$ for arbitrary large $\mathcal{T}$. The details of proof are left to Appendix B.

In the following theorem, we provide an example demonstrating that when two players use the optimistic gradient descent ascent or negative momentum method, their strategy will diverge at an exponential rate, regardless of how they choose their step-sizes and momentum parameters.

**Theorem 3.2.** *Consider a periodic game with period $\mathcal{T} = 2$, and described by the following payoff matrix*

$$A_t = \begin{cases} [1, -1], & t \text{ is odd} \\ [-1, 1], & t \text{ is even} \end{cases} \tag{6}$$

*with $x_t \in \mathbb{R}$, $y_t \in \mathbb{R}^2$. If two players use optimistic gradient descent ascent or negative momentum method, then regardless of how they choose step sizes and momentum parameters, we have*

$$\sup_{s \in [t]} \Delta_{i,s} \in \Omega(\lambda^t), \text{ where } \lambda > 1, \ i \in \{1, 2\}.$$

*Here $\lambda$ is determined by the largest modulus of the eigenvalues of the iterative matrix of optimistic gradient descent ascent or negative momentum method.*

To prove theorem 3.2, we directly calculate the characteristic polynomials of the iterative matrices products in one period for optimistic gradient descent ascent and negative momentum method under the game defined by (6). To show that these systems have an exponential divergence rate, it is sufficient to demonstrate that their characteristic polynomials have a root with modulus larger than 1. We achieve this by using the Schur stable theorem, which is also employed to demonstrate the last iterate convergence of several learning dynamics in time-independent game (Zhang and Yu [2020]). The proof is deferred to Appendix C.

In Figure (1), we present the function curves of $\Delta_{1,t}$ for these three types of game dynamics under the periodic game defined by (6). From the experimental results, extra-gradient converges, while both optimistic gradient descent ascent and negative momentum method diverge.

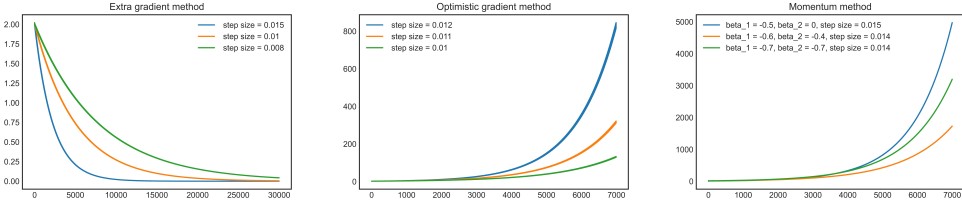

**Figure 1:** Function curves of $\Delta_{1,t}$ of the game presented in Theorem 3.2. Extra-gradient converges, while the other two methods diverge.

### 3.3 Convergent perturbed game

Recall that the payoff matrix of a convergent perturbed game has form $A_t = A + B_t$, where $\lim_{t \to \infty} B_t = 0$, and we refer to the zero-sum game defined by payoff matrix $A$ as the stable game. We denote

$$\Delta_t = \|A^\top x_t\|_2 + \|A y_t\|_2, \tag{7}$$

thus $\Delta_t$ measures the distance between the strategy $(x_t, y_t)$ in the $t$-th round and the Nash equilibrium of the stable game defined by $A$. Moreover, $\Delta_t = 0$ if and only if $(x_t, y_t)$ is an equilibrium of the stable game.

In the literature on linear difference systems, a common assumption that needs to be added for convergence guarantee is the following bounded accumulated perturbations (BAP) assumption (Benzaid and Lutz [1987], Elaydi et al. [1999], Elaydi and Györi [1995]):

$$\sum_{t=0}^{\infty} \|B_t\|_2 \text{ is bounded.} \qquad \text{(BAP assumption)}$$

In the following theorem, we prove that under BAP assumption, all three learning dynamics considered in this paper will make $\Delta_t$ converge to 0, with a rate dependent on the vanishing rate of $B_t$.

**Theorem 3.3.** *Assume that the* (BAP assumption) *holds, i.e., $\sum_{t=0}^{\infty}\|B_t\|_2$ is bounded, and let $\sigma$ be the maximum modulus of the singular value of payoff matrix $A$, then with parameters choice:*

- *for extra-gradient with step size $\alpha = \eta < \frac{1}{2\sigma}$,*

- *for optimistic gradient descent ascent with step size $\eta < \frac{1}{2\sigma}$,*

- *for negative momentum method with step size $\eta < \frac{1}{\sigma}$ and momentum parameters $\beta_1 = -\frac{1}{2}$ and $\beta_2 = 0$,*

*we have $\Delta_t$ converge to 0 with rate $\mathcal{O}(f(t))$. Here*

$$f(t) = \max\{\lambda^t, \sum_{i=t/2}^{\infty}\|B_i\|_2\},$$

*and $\lambda \in (0,1)$ is determined by the eigenvalues of the iterative matrix of corresponding learning dynamics and the payoff matrix $A$ of the stable game.*

There are two main ingredients in the proof of Theorem 3.3: firstly, we show that the iterative matrices associated with these learning dynamics are diagonalizable; secondly, these matrices do not have eigenvalues with modulus larger than 1. Moreover, we prove a general results which states any linear difference system satisfying these two conditions will converge. The details of proof are left to Appendix E.

**Remark 3.2.** *The* (BAP assumption) *can be converted into a constraint on the vanishing rate of $B_t$: if $B_t$ has a vanishing rate like $\mathcal{O}(\frac{1}{t^{1+\epsilon}})$, for some arbitrary small $\epsilon > 0$, then $\sum_{t=0}^{\infty}\|B_t\|_2$ is bounded. We also note that the condition for $B_t$ to vanish at a rate $\mathcal{O}(\frac{1}{t^{1+\epsilon}})$, $\forall \epsilon > 0$ is necessary to ensure convergence in general linear difference system. For example, consider the 1-dimension system $x_t = (1 + \frac{1}{t})x_{t-1}$ where a $\mathcal{O}(\frac{1}{t})$ convergence rate of perturbations leads to $x_t$ diverging with a $\Omega(t)$ rate.*

Surprisingly, in the next theorem, we prove that Extra-gradient makes $\Delta_t$ asymptotically converge to 0, without making any further assumptions about the converge rate of $B_t$.

**Theorem 3.4.** *In a convergent perturbed game, if two players use Extra-gradient, there holds $\lim_{t\to\infty} \Delta_t = 0$ with step size $\alpha = \eta < \frac{1}{2\sigma}$ where $\sigma$ is the maximum modulus of the singular value of payoff matrix $A$.*

To prove Theorem 3.4, we first observe that $\|x_t\|_2 + \|y_t\|_2$ a non-increasing function of $t$ since the iterative matrix of extra-gradient is a normal matrix. Next, we prove that if Theorem 3.4 doesn't hold, then $\|x_t\|_2 + \|y_t\|_2$ will decrease by a constant for infinite number of times, thus leading to a contradiction with the non-increasing property and the non-negativity of $\|x_t\|_2 + \|y_t\|_2$. The proof is deferred to Appendix F.

## 4 Experiments

In this section we present numerical results for our theoretical results in Section 3.

**Experiments on Theorem 3.1** We verify Theorem 3.1 through a period-3 game, the payoff matrices are chosen to be

$$A_1 = \begin{bmatrix} 1 & 2 \\ 2 & 4 \end{bmatrix}, A_2 = \begin{bmatrix} 3 & 7 \\ 7 & 1 \end{bmatrix}, A_3 = \begin{bmatrix} 4 & 2 \\ 4 & 2 \end{bmatrix}.$$

We run extra-gradient and optimistic gradient descent ascent on this example, both with step size = 0.01, the experimental results are presented in Figure (2). We can see extra-gradient (left) makes $\Delta_{i,t}$ converge, while optimistic gradient descent ascent (right) makes $\Delta_{i,t}$ diverge. This result supports Theorem 3.1 and provides a numerical example of the separation of extra-gradient and optimistic gradient descent ascent in periodic games.

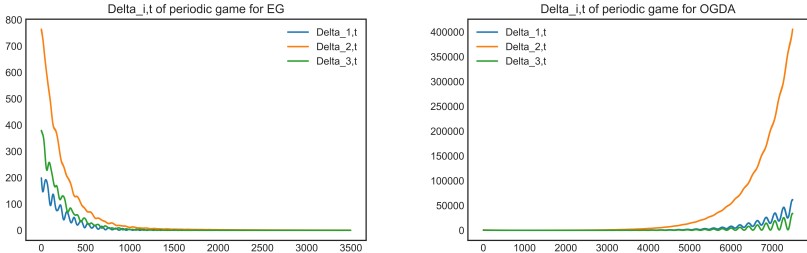

**Figure 2:** Function curves of $\Delta_t$ for extra-gradient (left), and optimistic gradient descent ascent (right).

**Experiments on Theorem 3.3** We verify Theorem 3.3 by examples:

$$A = \begin{bmatrix} 2 & 3 \\ 4 & 6 \end{bmatrix}, B_{1,t} = B \cdot t^{-1.1}, B_{2,t} = B \cdot t^{-4}, \text{ and }, B_{3,t} = B \cdot t^{-8}$$

where $B = [[-15, 70], [-90, 90]]$. The step size is chosen to be $0.005$. The initial points are chosen to be $x_0 = [15, 13], x_{-1} = [11, 3]$ and $y_0 = [35, 1], y_{-1} = [35, 1]$. The experimental results are presented in Figure 3, all of the three dynamics make $\Delta_t$ converge to 0, and slower convergence rate of perturbations can decelerate the convergence rate of learning dynamics, thus support the convergence result in Theorem 3.3. We also observe that the convergence rate is faster than the upper bound provided in the theorem. Therefore, we conjecture the bound of convergence rate can be improved.

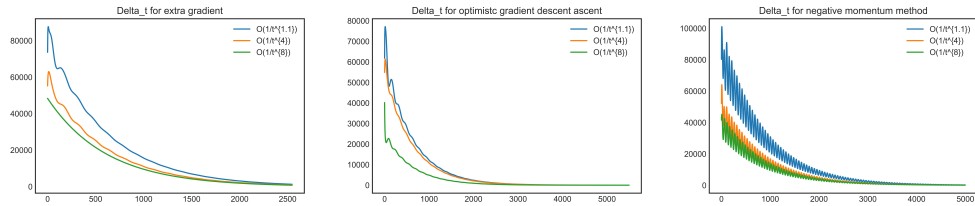

**Figure 3:** Values of $\Delta_t$ for extra-gradient (left), optimistic gradient descent ascent (middle), negative momentum method (right).

**Experiments on Theorem 3.4** We verify Theorem 3.4 by two group of examples. The perturbations are :

$$B_{1,t} = B \cdot t^{-0.4}, \ B_{2,t} = B \cdot t^{-0.3}, \ B_{3,t} = B \cdot t^{-0.2},$$

and

$$B_{4,t} = B \cdot \log(t)^{-1.8}, \ B_{5,t} = B \cdot \log(t)^{-1.5}, \ B_{6,t} = B \cdot \log(t)^{-1.3}.$$

where $B = [[-10, 10], [-10, 10]]$. The payoff matrix of stable game is chosen to be $A = [[2, 3], [4, 6]]$. Note that the perturbations do not satisfy (BAP assumption). The experimental results are shown in Figure (4). We can see all these curves converge to 0, thus support the result in Theorem 3.4. Furthermore, we can observe that large perturbations may lead to more oscillations during the convergence processes, which in turn slows down the rate of convergence. We present more experiments in Appendix G.

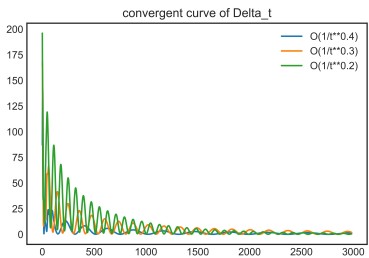 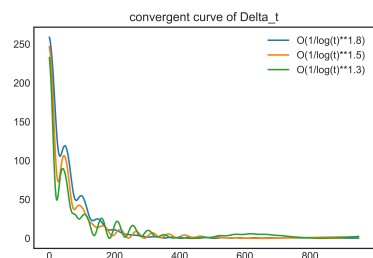

**Figure 4:** Values of $\Delta_t$ for extra-gradient with $A + B_{i,t}, i = 1, 2, 3$ (left), and $A_2 + B_{i,t}, i = 4, 5, 6$ (right).

## 5 Discussion

In this paper, we study the last-iterate behavior of extra-gradient, optimistic gradient descent ascent, and negative momentum method in two types of time-varying games : periodic game and convergent perturbed game. In the case of periodic game, we prove that extra-gradient will converge to a Nash equilibrium while other two methods diverge. To the best of our knowledge, this is the first result that provides a clear separation between the behavior of extra-gradient methods and optimistic gradient descent ascent. In the case of convergent perturbed game, we prove all three learning dynamics converge to Nash equilibrium under the BAP assumption, which is commonly used in the literature on dynamical systems.

Our results leave many interesting open questions. Firstly, is the BAP assumption necessary for ensuring convergence in optimistic gradient descent ascent and negative momentum method? Secondly, the bound of convergence rate in Theorem 3.3 may rather slight as we shown in experiments section. Obtaining a tighter bound on the convergence rate is an important future research problem. Thirdly, can the results here be generalized to other settings, such as constrained zero-sum game?

## Acknowledgement

Yi Feng is supported by the Fundamental Research Funds for the Central Universities. Ioannis Panageas wants to thank a startup grant. Xiao Wang acknowledges Grant 202110458 from SUFE and support from the Shanghai Research Center for Data Science and Decision Technology.

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
