# A Step sizes and eigenvalues of the iterative matrix

The eigenvalues of the iterative matrices in the linear differences systems in (2) (3) and (4) play a crucial role in analyzing the dynamic behavior of learning dynamics. In this section, we study how the choice of step sizes in learning dynamics affects the eigenvalues of these iterative matrices.

We firstly present the following corollary of Schur's theorem, which was also used in Zhang and Yu [2020] to demonstrate the convergence of learning dynamics in time-independent games.

**Lemma A.1.** *(Corollary 2.1 in Zhang and Yu [2020]). The roots of a real quartic polynomial $\lambda^4 + a\lambda^3 + b\lambda^2 + c\lambda + d$ are within the (open) unit disk of the complex plane if and only if $|c - ad| < 1 - d^2$, $|a + c| < b + d + 1$ and $b < (1 + d) + (c - ad)(a - c)/(d - 1)^2$.*

**Lemma A.2.** *Let $\sigma$ be the maximum modulus of the singular value of payoff matrix $A$. Then if for extra-gradient method with step size $\alpha = \gamma < \frac{1}{2\sigma}$, optimistic gradient descent ascent with step size $\eta < \frac{1}{2\sigma}$, and negative momentum method with step size $\eta < \frac{1}{\sigma}$ and momentum parameters $\beta_1 = -\frac{1}{2}$ and $\beta_2 = 0$, then for the iterative matrices $\mathcal{A}$ in (2) (3) and (4), we have the following conclusion:*

- *If payoff matrix $A$ is non-singular, then the modulus of eigenvalues of these iterative matrices $\mathcal{A}$ are strictly less than 1.*

- *If payoff matrix $A$ is singular, then 1 is an eigenvalue of the iterative matrix $\mathcal{A}$, and other eigenvalues of $\mathcal{A}$ have modulus strictly less than 1.*

*Proof.* **OGDA.** We first write the characteristic polynomials of the iterative matrix $\mathcal{A}$ in (2) when payoff matrix is equal to $A$. Recall in this case, we have

$$
\mathcal{A} = \begin{bmatrix} I & -2\eta A^\top & 0 & \eta A^\top \\ 2\eta A & I & -\eta A & 0 \\ I & 0 & 0 & 0 \\ 0 & I & 0 & 0 \end{bmatrix}. \tag{8}
$$

The characteristic polynomial equations are:

$$
\lambda^2 (\lambda - 1)^2 + \eta^2 \sigma_i^2 (1 - 2\lambda)^2 = 0, i \in [m] \tag{9}
$$

where $\sigma_i$ is a singular value of $A$. And then according to Lemma A.1, it is easy to verify if $0 < \eta\sigma < \frac{1}{2}$, then the norm of roots of the above polynomial is always less than 1. When $\sigma_i = 0$, we have the eigenvalues of $\mathcal{A}$ come from (9) are equal to 1.

In all, if the payoff matrix $A$ is non-singular, we have the modulus of eigenvalue of $\mathcal{A}$ is strictly smaller than 1. And if there exists some singular value of $A$ equals to 0, we can obtain that if $\sigma_i = 0$, then $\mathcal{A}$ has eigenvalue equal to 1, otherwise, $\mathcal{A}$ only has eigenvalues whose norm is less than 1.

**EG.** We first write characteristic polynomial of iterative matrix $\mathcal{A}$ in (3), with payoff matrix equals to $A$. We have

$$
\mathcal{A} = \begin{bmatrix} I - \alpha\gamma AA^\top & -\alpha A \\ \alpha A^\top & I - \gamma\alpha A^\top A \end{bmatrix}.
$$

The characteristic polynomial equations are:

$$
(\lambda - 1)^2 + 2\gamma\alpha\sigma_i^2(\lambda - 1) + \alpha^2\sigma_i^2 + \alpha^2\gamma^2\sigma_i^4 = 0, i \in [m]
$$

where $\sigma_i$ is a singular value of $A$. And then by Lemma A.1, the norm of roots of the above polynomial is always less than 1 if the following holds for all $i \in [m]$,

$$
\alpha^2\sigma_i^2 + (\alpha\gamma\sigma_i^2 - 1)^2 < 1. \tag{10}
$$

It is easy to verify that $\alpha = \gamma < \frac{1}{2\sigma}$ satisfies the above inequalities. Then we can use similar analysis in the part of OGDA to prove the conclusion for EG.

**Negative Momentum Method.** First we write characteristic polynomial of iterative matrix $\mathcal{A}$ defined in (4) when payoff payoff equals to $A$, we have

$$
\mathcal{A} = \begin{bmatrix}
(1+\beta_1)I & -\eta A & -\beta_1 I & 0 \\
\eta(1+\beta_1)A^\top & (1+\beta_2)I - \eta^2 A^\top A & -\eta\beta_1 A^\top & -\beta_2 I \\
I & 0 & 0 & 0 \\
0 & I & 0 & 0
\end{bmatrix}. \tag{11}
$$

The characteristic polynomial equations are:

$$
(\lambda - 1)^2(\lambda - \beta_1)(\lambda - \beta_2) + \eta^2 \sigma_i^2 \lambda^3 = 0, i \in [m]
$$

when $\beta_1 = -\frac{1}{2}$, $\beta_2 = 0$ and $\eta < \frac{1}{\sigma}$ satisfies conditions in Lemma A.1. We can also use a similar analysis as in the OGDA part to prove the conclusion for negative momentum method. □

## B  Omitted Proofs from Theorem 3.1

**Theorem 3.1.** *When two players use extra-gradient in a periodic games with period $T$, with step size $\alpha = \gamma < \frac{1}{\sigma}$ where $\sigma = \max\{\sigma' | \sigma'$ is a singular value of $A_i$ for some $i \in [\mathcal{T}]\}$. Then*

$$
\Delta_{i,t} \in \mathcal{O}\left((\lambda_*)^{t/\mathcal{T}} \cdot \mathrm{Poly}(t)\right), \ \forall i \in [\mathcal{T}]
$$

*where $\lambda_* = \max\{ |\lambda| \mid \lambda$ is an eigenvalue of $\left(\prod_{t=1}^{\mathcal{T}} \mathcal{A}_t\right), \lambda \neq 1\}$, and $\lambda_* < 1$.*

In this section, we prove Theorem 3.1. In the following, we use $\tilde{\mathcal{A}}$ to denote matrix $\prod_{i=1}^{T} \mathcal{A}_i$. As shown by the Floquet theorem, the asymptotic behavior of a periodic linear system is determined by the product of iterative matrices over one period. Therefore, the analysis can be reduced to that of an autonomous system. We analyze the Jordan normal form of the product matrix for extra-gradient. We prove that the product matrix has no eigenvalues with a modulus larger than 1. Moreover, the Jordan blocks of 1 as an eigenvalue of the product matrix have size equals to 1. These facts are enough to show the exponentially convergent behavior of extra-gradient. Before going through details of the proof, we provide a road map for the proof in Figure 5.

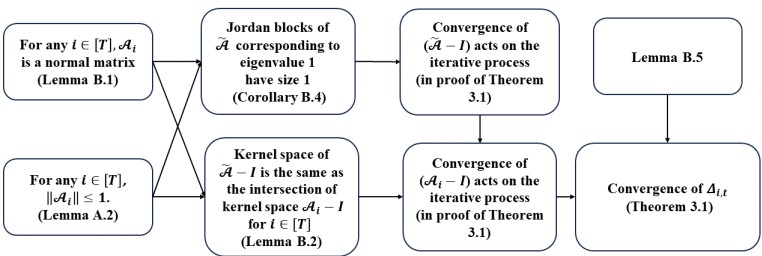

**Figure 5:** Road map for the proof of Theorem 3.1

Recall that EG can be written in a single linear difference system as

$$
\begin{bmatrix} x_{t+1} \\ y_{t+1} \end{bmatrix} = \begin{bmatrix} I - \alpha\gamma A_t A_t^\top & -\alpha A_t \\ \alpha A_t^\top & I - \gamma\alpha A_t^\top A_t \end{bmatrix} \begin{bmatrix} x_t \\ y_t \end{bmatrix}. \tag{12}
$$

Denote $\mathcal{A}_t$ the iterative matrix in (12). The following lemma tells us that $\mathcal{A}_t$ is a normal matrix.

**Lemma B.1.** *For any $i \in [T]$, $\mathcal{A}_i$ is a normal matrix.*

*Proof.* We have

$$\mathcal{A}_i \mathcal{A}_i^\top = \mathcal{A}_i^\top \mathcal{A}_i = \begin{bmatrix} (I - \alpha\gamma A_i A_i^\top)^2 + \alpha^2 A_i A_i^\top & 0 \\ 0 & (I - \alpha\gamma A_i^\top A_i)^2 + \alpha^2 A_i^\top A_i \end{bmatrix}.$$

$\square$

Using above lemma, we can present several useful lemmas to describe Jordan form of matrix $\tilde{\mathcal{A}}$.

**Lemma B.2.** *If $\alpha = \gamma < \frac{1}{\sigma}$, then for any $i \in [T]$, $\|\mathcal{A}_i\|_2 \leq 1$, and $\ker(\tilde{\mathcal{A}} - I) = \cap_{i=1}^T \ker(\mathcal{A}_i - I)$. Moreover, denote*

$$\lambda_* = \max\{ |\lambda| \mid \lambda \text{ is an eigenvalue of } \tilde{\mathcal{A}}, \lambda \neq 1 \},$$

*then we have $\lambda_* < 1$.*

*Proof.* $(\Leftarrow)$ : If $v \in \cap_{i=1}^T \ker(\mathcal{A}_i - I)$, then for any $i \in \{1, ..., T\}$, $\mathcal{A}_i v = v$, thus

$$\tilde{\mathcal{A}} v = \mathcal{A}_T \mathcal{A}_{T-1} ... \mathcal{A}_1 v = v.$$

Then we have $\cap_{i=1}^T \ker(\mathcal{A}_i - I) \subseteq \ker(\tilde{\mathcal{A}} - I)$.

$(\Rightarrow)$ : Let $v \in \ker(\tilde{\mathcal{A}} - I)$, then we have $\|v\|_2 = \|\mathcal{A}_T ... \mathcal{A}_1 v\|_2$. Denote $\|\cdot\|_2$ as 2-norm of matrices and vectors. According to Lemma A.2, if $\alpha = \gamma < \frac{1}{\sigma}$, then the spectral radius $\rho(\mathcal{A}_i)$ of $\mathcal{A}_i$ is no larger than 1. Combining with the fact that $\mathcal{A}_i$ is normal, we have $\|\mathcal{A}_i\|_2 = \rho(\mathcal{A}_i) \leq 1$ for $i \in [T]$. We claim that if $\|v\|_2 = \|\mathcal{A}_T ... \mathcal{A}_1 v\|_2$, then we have $\mathcal{A}_i v = v$ for $i \in [T]$. We prove it by contradiction. Suppose the claim is not true. Let $s$ be the minimum $i$ such that $\mathcal{A}_i v \neq v$. Since $\mathcal{A}_s$ is normal and its eigenvalues whose modulus equal to 1 can only be 1, we have $\|\mathcal{A}_s v\|_2 < \|v\|_2$, then there holds

$$\begin{aligned} \|v\|_2 &= \|\mathcal{A}_T ... \mathcal{A}_s ... \mathcal{A}_1 v\|_2 \\ &= \|\mathcal{A}_T ... \mathcal{A}_s v\|_2 \\ &< \|\mathcal{A}_T ... \mathcal{A}_{s+1}\|_2 \|v\|_2 \\ &\leq \|v\|_2, \end{aligned}$$

which leads to a contradiction. Therefore, for any $i \in [T]$, we obtain that $\mathcal{A}_i v = v$, i.e., $v \in \ker(\mathcal{A}_i - I)$. From the claim, we know that if $v \in \ker(\tilde{\mathcal{A}} - I)$ , then $v \in \ker(\mathcal{A}_i - I)$ for $i \in [T]$. Thus we have $\ker(\tilde{\mathcal{A}} - I) \subseteq \cap_{i=1}^T \ker(\mathcal{A}_i - I)$.

Next we prove that $\lambda_* \leq 1$. By the definition of $\lambda_*$, we obtain

$$\begin{aligned} \lambda_* &\leq \rho(\mathcal{A}_T \cdots \mathcal{A}_1) \\ &\leq \|\mathcal{A}_T \cdots \mathcal{A}_1\|_2 \\ &\leq \|\mathcal{A}_T\|_2 \cdots \|\mathcal{A}_1\|_2 \leq 1, \end{aligned}$$

where the second inequality holds because the spectral radius $\rho(A) \leq \|A\|_2$ for any matrix $A$.

Now we prove that $\lambda_* \neq 1$, which means that $\tilde{\mathcal{A}}$ have no eigenvalue $\lambda$ satisfying $\lambda \neq 1$ and $|\lambda| = 1$. Assuming $v$ is the eigenvector of $\tilde{\mathcal{A}}$ corresponding to $\lambda'$, where $|\lambda'| = 1$, we can obtain $\|\mathcal{A}_T ... \mathcal{A}_1 v\|_2 = \|\lambda' v\|_2 = \|v\|_2$. Similar to the proof above, $\mathcal{A}_i v = v$ for $i \in [T]$, which implies that $\lambda' = 1$. This completes the proof of $\lambda_* < 1$.

$\square$

**Lemma B.3.** *Under a suitable orthogonal normal basis, $\tilde{\mathcal{A}}$ has form*

$$\begin{bmatrix} \mathbf{I}_{\mathbf{r} \times \mathbf{r}} & \mathbf{0} \\ \mathbf{0} & \mathbf{C} \end{bmatrix}, \tag{13}$$

*where $\mathbf{I}_{\mathbf{r} \times \mathbf{r}} \in \mathbb{R}^{\mathbf{r} \times \mathbf{r}}$, $\mathbf{C} \in \mathbb{R}^{(\mathbf{n+m-r}) \times (\mathbf{n+m-r})}$, and $r = \dim_{\mathbb{R}}(\ker(\tilde{\mathcal{A}} - I))$.*

*Proof.* Let $\{v_1, ..., v_r\}$ be an orthogonal normal basis of $\ker(\tilde{\mathcal{A}} - I)$, i.e., $\langle v_i, v_j \rangle = 1$ if $i = j$, $\langle v_i, v_j \rangle = 0$ if $i \neq j$. First, we extend $\{v_1, ..., v_r\}$ to an orthonormal basis of $\mathbb{R}^{n+m}$ and denote this basis by $\{v_1, ..., v_r, v_{r+1}, ..., v_{n+m}\}$. We also denote $M$ the matrix consisting of $\{v_1, ..., v_r, v_{r+1}, ..., v_{n+m}\}$ as columns. With these settings, we have $M^\top M = MM^\top = I$.

Under this basis, $\mathcal{A}_i$ is represented by matrix

$$\begin{bmatrix} \mathbf{I_{r \times r}} & \mathbf{0} \\ \mathbf{C_{i,1}} & \mathbf{C_{i,2}} \end{bmatrix}. \tag{14}$$

Moreover, as $\mathcal{A}_i$ is a normal matrix, its representation under an orthogonal normal basis is still a normal matrix, thus we have

$$\begin{bmatrix} \mathbf{I_{r \times r}} & \mathbf{0} \\ \mathbf{C_{i,1}} & \mathbf{C_{i,2}} \end{bmatrix} \begin{bmatrix} \mathbf{I_{r \times r}} & \mathbf{C_{i,1}^\top} \\ \mathbf{0} & \mathbf{C_{i,2}^\top} \end{bmatrix} = \begin{bmatrix} \mathbf{I_{r \times r}} & \mathbf{C_{i,1}^\top} \\ \mathbf{0} & \mathbf{C_{i,2}^\top} \end{bmatrix} \begin{bmatrix} \mathbf{I_{r \times r}} & \mathbf{0} \\ \mathbf{C_{i,1}} & \mathbf{C_{i,2}} \end{bmatrix}. \tag{15}$$

Note that (15) is equivalent to

$$\begin{bmatrix} \mathbf{I_{r \times r}} & \mathbf{C_{i,1}^\top} \\ \mathbf{C_{i,1}} & \mathbf{C_{i,1}C_{i,1}^\top} + \mathbf{C_{i,2}C_{i,2}^\top} \end{bmatrix} = \begin{bmatrix} \mathbf{I_{r \times r}} + \mathbf{C_{i,1}^\top C_{i,1}} & \mathbf{C_{i,1}^\top C_{i,2}} \\ \mathbf{C_{i,2}^\top C_{i,1}} & \mathbf{C_{i,2}^\top C_{i,2}} \end{bmatrix}.$$

As a consequence, we have $\mathbf{C_{i,1}^\top C_{i,1}} = \mathbf{0}$, and furthermore, this implies $\mathbf{C_{i,1}} = \mathbf{0}$. Thus (14) has form

$$\begin{bmatrix} \mathbf{I_{r \times r}} & \mathbf{0} \\ \mathbf{0} & \mathbf{C_{i,2}} \end{bmatrix},$$

and under this basis, $\tilde{\mathcal{A}}$ can be represented by

$$\begin{bmatrix} \mathbf{I_{r \times r}} & \mathbf{0} \\ \mathbf{0} & \prod_{i=1}^T \mathbf{C_{i,2}} \end{bmatrix}.$$

Since $\prod_{i=1}^T \mathbf{C_{i,2}}$ is a matrix with size of $(n + m - r) \times (n + m - r)$, we complete the proof. $\square$

**Corollary B.4.** *If $\lambda = 1$ is an eigenvalue of $\tilde{\mathcal{A}}$, then the Jordan blocks of $\tilde{\mathcal{A}}$ corresponding to eigenvalue $1$ has size $1$.*

*Proof.* From Lemma B.3, we have a decomposition $\mathbb{R}^{m+n} = \ker(\tilde{\mathcal{A}} - I) \oplus V'$, and both these two spaces are invariant under the action of $\tilde{\mathcal{A}}$. Thus we can choose a basis of $V'$ consisting of Jordan chains and denote this basis by $\{w_1, ..., w_{m+n-r}\}$, then under the basis $\{w_1, ..., w_{m+n-r}\} \cup \{v_1, ..., v_r\}$, $\tilde{\mathcal{A}}$ is a block diagonal matrix. Moreover, there is no eigenvectors corresponding to eigenvalue $1$ in $\{w_1, ..., w_{m+n-r}\}$, because any $w_i$ is linearly independent with $\{v_1, ..., v_r\}$ (since they are basis), thus if some $w_i$ is an eigenvector of eigenvalue $1$, then a contradiction is conducted since it is assumed that $\dim_{\mathbb{R}}(\ker(\tilde{\mathcal{A}} - I)) = r$. $\square$

**Lemma B.5.** *Denote $X_t = (x_t, y_t)$ be the strategies of players at round of $t$ when they are playing extra-gradient. For any $i \in [T]$, if $\|(\mathcal{A}_i - I)X_t\|_2$ converges to $0$ with rate $\mathcal{O}\left((\lambda_*)^{t/T} \cdot \mathrm{Poly}(t)\right)$, then $\Delta_{i,t}$ converges to $0$ with rate $\mathcal{O}\left((\lambda_*)^{t/T} \cdot \mathrm{Poly}(t)\right)$.*

*Proof.* Writing $(\mathcal{A}_i - I)X_t$ in a matrix form:

$$\begin{bmatrix} -\alpha\gamma A_i A_i^\top & -\alpha A_i \\ \alpha A_i^\top & -\alpha\gamma A_i^\top A_i \end{bmatrix} \begin{bmatrix} x_{t-1} \\ y_{t-1} \end{bmatrix} = \begin{bmatrix} -\alpha\gamma A_i A_i^\top x_{t-1} - \alpha A_i y_{t-1} \\ \alpha A_i^\top x_{t-1} - \alpha\gamma A_i^\top A_i y_{t-1} \end{bmatrix}.$$

For the sake of readability, we denote $g(t) = (\lambda_*)^{t/T} \cdot \mathrm{Poly}(t)$. According to the assumption, there is a constant $c$ such that $\|(\mathcal{A}_i - I)X_t\|_2 \le cg(t)$, then we have

$$\|-\gamma A_i A_i^\top x_{t-1} - A_i y_{t-1}\|_2 \le \frac{cg(t)}{\alpha},$$

$$\|A_i^\top x_{t-1} - \gamma A_i^\top A_i y_{t-1}\|_2 \le \frac{cg(t)}{\alpha}.$$

Let $c_1 = \max\{\|A_i\|_2, i \in [T]\}$. Using these two inequalities to bound $\|A_i^\top x_t\|_2$, we have

$$\|(\gamma^2 A_i^\top A_i + I)A_i^\top x_{t-1}\|_2$$

$$= \|\gamma^2 A_i^\top A_i A_i^\top x_{t-1} + A_i^\top x_{t-1}\|_2$$

$$= \|A_i^\top x_{t-1} - \gamma A_i^\top A_i y_{t-1} - \gamma A_i^\top(-\gamma A_i A_i^\top x_{t-1} - A_i y_{t-1})\|_2$$

$$\le \|A_i^\top x_{t-1} - \gamma A_i^\top A_i y_{t-1}\|_2 + \gamma\|A_i^\top\|_2\|-\gamma A_i A_i^\top x_{t-1} - A_i y_{t-1}\|_2$$

$$\le \frac{c(1 + \gamma c_1)g(t)}{\alpha}.$$

Since matrix $\gamma^2 A_i^\top A_i + I$ is invertible, then

$$\|A_i x_t\|_2 = (\gamma^2 A_i^\top A_i + I)^{-1}(\gamma^2 A_i^\top A_i + I)A_i^\top x_{t-1}\|_2$$

$$\le \|(\gamma^2 A_i^\top A_i + I)^{-1}\|_2\|(\gamma^2 A_i^\top A_i + I)A_i^\top x_{t-1}\|_2$$

$$\le \|(\gamma^2 A_i^\top A_i + I)A_i^\top x_{t-1}\|_2$$

$$\le \frac{c(1 + \gamma)g(t)}{\alpha},$$

where the last inequality is due to $\|(\gamma^2 A_i^\top A_i + I)^{-1}\|_2 \le 1$. Similarly, we can obtain

$$\|A_i y_t\|_2 \le \frac{c(1 + \gamma c_1)g(t)}{\alpha}.$$

Thus by definition of $\Delta_{i,t} = \|A_i^\top x_t\|_2 + \|A_i y_t\|_2$, $\Delta_{i,t}$ converges to 0 with rate $\mathcal{O}\left((\lambda_*)^{t/T} \cdot \mathrm{Poly}(t)\right)$. $\qquad\square$

Now we are ready to prove Theorem 3.1.

*Proof of Theorem 3.1.* We have proved $\lambda_* < 1$ in Lemma B.2, now we prove the part of convergence rate. Note that here we cannot directly apply the Floquet theorem in Proposition 2.4, as it requires all iterative matrices within a period to be invertible. However, the proof here follows the same idea as the Floquet theorem : the convergence behavior of a periodic linear difference system is determined by the product of all iterative matrices of the system in a period. According to Corollary B.4, we can write Jordan form $J$ of $\tilde{\mathcal{A}}$ in the following way:

$$J = \begin{bmatrix} I & 0 \\ 0 & \tilde{J} \end{bmatrix},$$

where $\tilde{J}$ consists of Jordan blocks corresponding to eigenvalues whose modulus not equal to 1. According to Lemma B.2, we have that the modulus of eigenvalues of $\tilde{J}$ are less than 1. Moreover, we assume

$$J = P^{-1}\tilde{A}P.$$

Denote $J_k(\lambda)$ as a Jordan block corresponding to eigenvalue $\lambda$ with size $k$, and $|\lambda| < 1$. We can write $J_k(\lambda) = \lambda I + N$, where $N$ represents the nilpotent matrix whose superdiagonal contains 1's and all other entries are zero. Moreover, we have $N^k = 0$ and $\|N\|_2 = 1$.

For each Jordan block $J_k(\lambda)$, without loss of generality, when $s > 2k$, by the binomial theorem:

$$J_k^s(\lambda) = (\lambda I + N)^s = \sum_{r=0}^{s} \binom{s}{r} \lambda^{s-r} N^r.$$

Then

$$\|J_k^s(\lambda)\|_2 \leq (k-1)\binom{s}{k-1}|\lambda|^{s-k+1},$$

since $\|N\|_2 = 1$ and $s > 2k$. We know that $\binom{s}{k-1}$ is a polynomial of $s$ with degree $k \leq n + m$. Since $|\lambda| < 1$, $\|J_k^s(\lambda)\|_2$ goes to zero in rate $\mathcal{O}\left((\lambda_*)^s \cdot \mathrm{Poly}(s)\right)$. Since $J_k^s(\lambda)$ are blocks in block diagnol matrix $\tilde{J}^s$, then

$$\|\tilde{J}^s\|_2 \leq \sum_{\lambda \in \mathrm{Eigenvalue}(\tilde{A}), \lambda \neq 1} \|J_k^s(\lambda)\|_2,$$

and $\|\tilde{J}^s\|_2$ goes to zero in rate $\mathcal{O}\left((\lambda_*)^s \cdot \mathrm{Poly}(s)\right)$. For any $t$, without loss of generality, we assume that $t = sT + j$, and $j \in [T]$ is the remainder. Then we have

$$(\tilde{A} - I)X_t = (\tilde{A} - I)\tilde{A}^s X_j$$

$$= (\tilde{A}^{s+1} - \tilde{A}^s)X_j$$

$$= P^{-1}(J^{s+1} - J^s)PX_j$$

$$= P^{-1}\left(\begin{bmatrix} I & 0 \\ 0 & J_1^{s+1} \end{bmatrix} - \begin{bmatrix} I & 0 \\ 0 & J_1^s \end{bmatrix}\right)PX_j$$

$$= P^{-1}\left(\begin{bmatrix} 0 & 0 \\ 0 & J_1^{s+1} - J_1^s \end{bmatrix}\right)PX_j.$$

Taking norm on both sides, we have

$$\|(\tilde{A} - I)X_t\|_2 \leq (\|J_1^{s+1}\|_2 + \|J_1^s\|_2)\|X_j\|_2 \leq 2\|J_1^s\|_2\|X_j\|_2.$$

From definition of $s$, we know that $s = \lfloor t/T \rfloor \geq t/T - 1$, leading to $(\lambda_*)^s \leq \frac{1}{\lambda_*}(\lambda_*)^{t/T}$. Since $\|J_1^s\|_2$ converges to zero with rate $\mathcal{O}\left((\lambda_*)^s \cdot \mathrm{Poly}(s)\right)$, then $\|(\tilde{A} - I)X_t\|_2$ converges to zero with rate $\mathcal{O}\left((\lambda_*)^{t/T} \cdot \mathrm{Poly}(t)\right)$. By Lemma B.2, for any $i \in [T]$, $\|(\mathcal{A}_i - I)X_t\|_2$ goes to zero in rate $\mathcal{O}\left((\lambda_*)^{t/T} \cdot \mathrm{Poly}(t)\right)$.

According to Lemma B.5, we conclude for any $i \in [T]$, $\Delta_{i,t}$ goes to zero with convergence rate $\mathcal{O}\left((\lambda_*)^{t/T} \cdot \mathrm{Poly}(t)\right)$, this completes the proof. $\qquad\square$

# C Omitted Proofs from Theorem 3.2

**Theorem 3.2.** *Consider a periodic game with period $\mathcal{T} = 2$, and described by the following payoff matrix*

$$A_t = \begin{cases} [1, -1], & t \text{ is odd} \\ [-1, 1], & t \text{ is even} \end{cases} \tag{6}$$

*with $x_t \in \mathbb{R}$, $y_t \in \mathbb{R}^2$. If two players use optimistic gradient descent ascent or negative momentum method, then regardless of how they choose step sizes and momentum parameters, we have*

$$\sup_{s \in [t]} \Delta_{i,s} \in \Omega(\lambda^t), \text{ where } \lambda > 1, \ i \in \{1, 2\}.$$

*Here $\lambda$ is determined by the largest modulus of the eigenvalues of the iterative matrix of optimistic gradient descent ascent or negative momentum method.*

## C.1 On initialization

Before proving Theorem 3.2, we discuss a more detailed question :

*Which initial points will make (OGDA) and (NM) diverge ?*

In fact, it is obviously that not every initial point will make optimistic gradient descent ascent and negative momentum method diverge. For example, if the initial point is chosen to be

$$(x_0, y_0) \in (\ker A_t^\top, \ker A_t), \tag{16}$$

then these point will be not diverge because they are stationary points of the game dynamics.

In the proof of Theorem 3.2 below, we explicitly construct initial points that diverge exponentially fast under (OGDA) or (NM). In Figure 6, we present an example of an initial point that converges under (NM) with the game defined by (35). In fact, we can see that these converge initial points of (OGDA) or (NM) lie on a low dimension space, thus have measure zero. Note that this doesn't conflict with Theorem 3.2, since we are **not** claiming that optimistic gradient descent ascent or negative momentum method will make every initial point diverge.

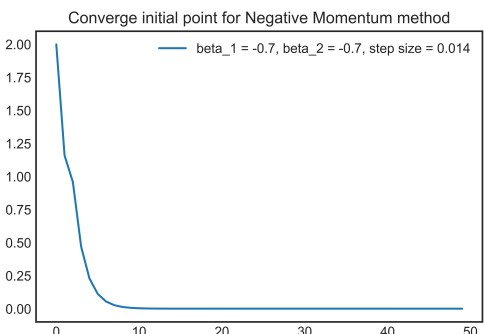

**Figure 6:** A converge initial point for negative momentum method, with initial condition $x_0 = x_{-1} = 0$, and $y_0 = (-0.4, 1)$, $y_{-1} = (1, -1)$. The curve is $\Delta_{1,t}$.

In the following sections C.2 and C.3, we will prove that both negative momentum method and OGDA diverge with an exponential rate under certain initial conditions. The proof idea is the same for these two learning dynamics : firstly, we prove that the product of iterative matrices in a period for these learning dynamics have an eigenvalue with modulus larger than 1; then, we show that eigenvectors corresponding to this eigenvalue as initial condition will diverge under the learning dynamics.

## C.2 Negative Momentum Method

We first consider negative momentum method with step size $\eta$, recall it can be written as:

$$x_{t+1} = x_t - \eta A_t y_t + \beta_1(x_t - x_{t-1}),$$
$$y_{t+1} = y_t + \eta A_{t+1}^\top x_{t+1} + \beta_2(y_t - y_{t-1}),$$

where $\beta_1, \beta_2 \le 0$ are the momentum parameters. Writing negative momentum method in matrix form, we have

$$
\begin{bmatrix} x_{t+1} \\ y_{t+1} \\ x_t \\ y_t \end{bmatrix} = \begin{bmatrix} (1+\beta_1)I & -\eta A_t & -\beta_1 I & 0 \\ \eta(1+\beta_1)A_{t+1}^\top & (1+\beta_2)I - \eta^2 A_{t+1}^\top A_t & -\eta\beta_1 A_{t+1}^\top & -\beta_2 I \\ I & 0 & 0 & 0 \\ 0 & I & 0 & 0 \end{bmatrix} \begin{bmatrix} x_t \\ y_t \\ x_{t-1} \\ y_{t-1} \end{bmatrix} \tag{17}
$$

Denote the iterative matrix in (17) as $\mathcal{A}_t$ and $X_t = (x_t^\top, y_t^\top, x_{t-1}^\top, y_{t-1}^\top)^\top$. Let $\tilde{\mathcal{A}}_{NM} = \mathcal{A}_{t+1}\mathcal{A}_t$, by Floquet Theorem, $\tilde{\mathcal{A}}_{NM}$ will determine the dynamical behaviors of negative momentum method. We have

$$
X_t = \tilde{\mathcal{A}}_{NM} X_{t-2}, \text{ for any } t \ge 2.
$$

In the following lemma, we show that the spectral radius of $\tilde{\mathcal{A}}_{NM}$ is always larger than 1.

**Lemma C.1.** *For any step size $\eta > 0$, and momentum parameters $\beta_1$, $\beta_2 \le 0$, the spectral radius of $\tilde{\mathcal{A}}_{NM}$ is larger than 1.*

*Proof.* We directly compute the characteristic polynomial $P_{\tilde{\mathcal{A}}_{NM}}(\lambda)$ of matrix $\tilde{\mathcal{A}}_{NM}$ as follows [‡]:

$$
P_{\tilde{\mathcal{A}}_{NM}}(\lambda) = \det(\lambda I - \tilde{\mathcal{A}}_{NM})
$$

$$
= [\lambda^4 - \left(4\eta^4 + 4\eta^2(\beta_2 - \beta_1) + \beta_1^2 + \beta_2^2 + 2\right) \cdot \lambda^3
$$

$$
+ \left(4\eta^2(\beta_2 - \beta_1) + \beta_1^2\beta_2^2 + 2\beta_1^2 + 2\beta_2^2 + 1\right) \cdot \lambda^2
$$

$$
- \left(2\beta_1^2\beta_2^2 + \beta_1^2 + \beta_2^2\right) \cdot \lambda + \beta_1^2\beta_2^2] \cdot (\lambda - 1) \cdot (\lambda - \beta_2^2).
$$

Note that this is a polynomial on $\lambda$ of degree 6, with two roots $\lambda = 1$ and $\lambda = \beta_2^2$.

Thus, eigenvalues of matrix $\tilde{\mathcal{A}}_{NM}$ consists of 1, $\beta_2^2$ and roots of the quartic polynomial

$$
g(\lambda) = \lambda^4 + a\lambda^3 + b\lambda^2 + c\lambda + d,
$$

with coefficients

$$
a = -(4\eta^4 + 4\eta^2(\beta_2 - \beta_1) + \beta_1^2 + \beta_2^2 + 2),
$$

$$
b = 4\eta^2(\beta_2 - \beta_1) + \beta_1^2\beta_2^2 + 2\beta_1^2 + 2\beta_2^2 + 1,
$$

$$
c = -(2\beta_1^2\beta_2^2 + \beta_1^2 + \beta_2^2),
$$

$$
d = \beta_1^2\beta_2^2.
$$

In order to prove the spectral radius of $\tilde{\mathcal{A}}_{NM}$ is larger than 1, we just need to verify that the maximal modulus of the roots of $g$ is larger than 1. According to Lemma A.1, the polynomial $g(\lambda)$ has a root with modulus no less than 1 if $|a + c| > b + d + 1$. Next we want to prove that $|a + c| > b + d + 1$ holds for any step size $\eta > 0$ and any momentum parameter $\beta_1, \beta_2$. Computing directly,

$$
\begin{aligned}
|a + c| - (b + d + 1) &= -(a + c) - (b + d + 1) \\
&= 4\eta^4 + 4\eta^2(\beta_2 - \beta_1) + \beta_1^2 + \beta_2^2 + 2 + 2\beta_1^2\beta_2^2 + \beta_1^2 + \beta_2^2 \\
&\quad - (4\eta^2(\beta_2 - \beta_1) + \beta_1^2\beta_2^2 + 2\beta_1^2 + 2\beta_2^2 + 1 + \beta_1^2\beta_2^2 + 1) \\
&= 4\eta^4 > 0,
\end{aligned}
$$

---

[‡]Symbolic computing software, such as Matlab, can be used to perform the computation of characteristic polynomial.

where the first equality holds since

$$a + c = -(4\eta^4 + 4\eta^2(\beta_2 - \beta_1) + 2\beta_1^2 + 2\beta_2^2 + 2 + 2\beta_1^2\beta_2^2)$$
$$\leq -(4\eta^4 + 4\eta^2(\beta_2 - \beta_1) + (\beta_1 - \beta_2)^2)$$
$$\leq -(2\eta^2 + \beta_2 - \beta_1)^2$$
$$\leq 0.$$

The inequality $|a + c| > b + d + 1$ violates the second condition in Corollary A.1, which means the maximal modulus of the roots of $g$ is at least 1.

Next, we want to prove the maximal modulus of the roots of $g$ is strictly larger than 1. Assuming that the maximal modulus of the roots of $g$ is equal to 1, then for any given $r > 1$, the roots of

$$(r\lambda)^4 + a(r\lambda)^3 + b(r\lambda)^2 + c(r\lambda) + d = 0$$

are within the (open) unit disk of the complex plane (the roots $\lambda$ satisfy $|r\lambda| \leq 1$). Divide the quartic polynomial by $r^4$, we have $\lambda^4 + \frac{a}{r}\lambda^3 + \frac{b}{r^2}\lambda^2 + \frac{c}{r^3}\lambda + \frac{d}{r^4} = 0$. By Corollary A.1, we have the second condition for the polynomial above, that is,

$$\left|\frac{a}{r} + \frac{c}{r^3}\right| < \frac{b}{r^2} + \frac{d}{r^4} + 1.$$

Notice that the inequality above holds for any $r > 1$. Let $r \to 1^+$, we obtain,

$$|a + c| \leq b + d + 1$$
$$(\Leftrightarrow)$$
$$|4\eta^4 + 4\eta^2(\beta_2 - \beta_1) + 2\beta_1^2 + 2\beta_2^2 + 2 + 2\beta_1^2\beta_2^2|$$
$$\leq 4\eta^2(\beta_2 - \beta_1) + 2\beta_1^2 + 2\beta_2^2 + 2\beta_1^2\beta_2^2 + 2$$
$$(\Leftrightarrow)$$
$$\eta^4 \leq 0.$$

which contradicts with the step size $\eta > 0$. Therefore, our assumption that the maximal modulus of the roots of $g$ is equal to 1 cannot hold.

In conclusion, we have that the spectral radius of $\tilde{\mathcal{A}}_{NM}$ is strictly greater than 1. $\qquad\square$

Now we are ready to proof Theorem 3.2 for the part of negative momentum method.

*proof of Theorem 3.2.* **(part I, Negative Momentum).** As we have shown in Lemma C.1, 1 and $\beta_2^2$ are two eigenvalues of $\tilde{\mathcal{A}}_{NM}$. We claim that if $[0, a, a, 0, b, b]^\top \in \mathbb{R}^6$ is an eigenvector of $\tilde{\mathcal{A}}_{NM}$, with condition $a$ and $b$ not simultaneously equal to 0, then it can only be an eigenvector corresponds to either 1 or $\beta_2^2$. In the following, we prove the above claim.

Without loss of generality, we assume that $b \neq 0$ (the case $a \neq 0$ is similar), moreover, we can assume that $b = 1$ by a normalization. Then, we have

$$\tilde{\mathcal{A}}_{NM} \cdot \begin{bmatrix} 0 \\ a \\ a \\ 0 \\ 1 \\ 1 \end{bmatrix} = \begin{bmatrix} 0 \\ a(\beta_2^2 + \beta_2 + 1) - \beta_2(\beta_2 + 1) \\ a(\beta_2^2 + \beta_2 + 1) - \beta_2(\beta_2 + 1) \\ 0 \\ a(\beta_2 + 1) - \beta_2 \\ a(\beta_2 + 1) - \beta_2 \end{bmatrix}$$

Firstly, if $a = 0$ and $[0, a, a, 0, 1, 1]^\top = [0, 0, 0, 0, 1, 1]^\top$ is an eigenvector of $\tilde{\mathcal{A}}_{NM}$, then either $\beta_2 = 0$ or $\beta_2 = -1$. If $\beta_2 = 0$, then $[0, 0, 0, 0, 1, 1]^\top$ is eigenvector corresponding to eigenvalue $\beta_2^2$; if $\beta_2 = -1$, then $[0, 0, 0, 0, 1, 1]^\top$ is eigenvector corresponding to eigenvalue 1.

Secondly, if $a \neq 0$ and $[0, a, a, 0, 1, 1]^\top$ is eigenvector of $\tilde{\mathcal{A}}_{NM}$, then

$$\frac{a(\beta_2^2 + \beta_2 + 1) - \beta_2(\beta_2 + 1)}{a} = \frac{a(\beta_2 + 1) - \beta_2}{1}$$
$$\Rightarrow (a - \beta_2)(a - 1)(\beta_2 + 1) = 0.$$

When $\beta_2 \neq -1$, then $a = \beta_2$ or $a = 1$, and $[0, a, a, 0, 1, 1]^\top$ is an eigenvector corresponding to eigenvalue $\beta_2^2$ or 1. When $\beta_2 = -1$, $[0, 1, 1, 0, 1, 1]^\top$ and $[0, 0, 0, 0, 1, 1]^\top$ are eigenvectors of $\tilde{\mathcal{A}}_{NM}$ corresponding to eigenvalue 1. Thus we conclude for any $a$ and $b$ not simultaneously equal to 0, $[0, a, a, 0, b, b]^\top$ can only be an eigenvector of $\tilde{\mathcal{A}}_{NM}$ corresponding to eigenvalue 1 and $\beta_2^2$, this completes the proof of the claim.

Next we construct an initial condition that has exponential divergence rate under negative momentum method. Let $\lambda'$ be the eigenvalue of $\tilde{\mathcal{A}}_{NM}$ with largest modulus except $\beta_2^2$, then by Lemma C.1, $|\lambda'| > 1$. We also denote $X_0 = [x_0, y_{0,1}, y_{0,2}, x_{-1}, y_{-1,1}, y_{-1,2}]^\top \in \mathbb{R}^6$ as the corresponding eigenvector of $\lambda'$. Here $y_i = (y_{i,1}, y_{i,2})$, and $x_i$ for $i = 0, -1$ are initial conditions. Then from the claim proved above, one of $x_0$, $x_{-1}$, $y_{0,1} - y_{0,2}$ and $y_{-1,1} - y_{-1,2}$ not equals to 0. Let $c = \max\{|x_0|, |x_{-1}|, |y_{0,1} - y_{0,2}|, |y_{-1,1} - y_{-1,2}|\}$, then $c > 0$.

In the following, we construct the initial point by considering two cases : $\lambda'$ is a real number or complex number.

Firstly, we consider the case that $\lambda'$ is a real number. We can write the iterative process using $\tilde{\mathcal{A}}_{NM}$ as follows:
$$X_{2t} = \tilde{\mathcal{A}}_{NM}^t X_0 = (\lambda')^t X_0.$$
which implies
$$x_{2t} = (\lambda')^t x_0, y_{2t,1} = (\lambda')^t y_{0,1}, y_{2t,2} = (\lambda')^t y_{0,2},$$
$$x_{2t-1} = (\lambda')^t x_{-1}, y_{2t-1,1} = (\lambda')^t y_{-1,1}, y_{2t-1,2} = (\lambda')^t y_{-1,2}.$$

Since $A_1 = [1, -1]$, then
$$A_1^\top x_t = [x_t, -x_t]^\top, A_1 y_t = y_{t,1} - y_{t,2},$$
$$\Rightarrow \Delta_{1,t} = \|A_1^\top x_t\|_2 + \|A_1 y_t\|_2$$
$$= \sqrt{2}|x_t| + |y_{t,1} - y_{t,2}|$$
$$= \begin{cases} |\lambda'|^{\frac{t}{2}}(\sqrt{2}|x_0| + |y_{0,1} - y_{0,2}|), & \text{if t is even} \\ |\lambda'|^{\frac{t+1}{2}}(\sqrt{2}|x_{-1}| + |y_{-1,1} - y_{-1,2}|), & \text{if t is odd.} \end{cases}$$

Then, $\max\{\Delta_{1,t-1}, \Delta_{1,t}\} \geq c|\lambda'|^{\frac{t}{2}}$. Let $\lambda = |\lambda'|^{\frac{1}{2}}$, we have $\sup_{s \in [t]} \Delta_{1,s} \geq c\lambda^t \in \Omega(\lambda^t)$. Similarly, then we have $\sup_{s \in [t]} \Delta_{2,s} \in \Omega(\lambda^t)$.

Secondly, we consider $\lambda'$ as a complex number. Denote this eigenvalue by $a + bi$, then $a - bi$ is also an eigenvalue of $\tilde{\mathcal{A}}_{NM}$. Denote $v$ the eigenvector of eigenvalue $a + bi$, then $\bar{v}$ is the eigenvector of eigenvalue $a - bi$. Let $X_0 = v + \bar{v}$. In the following, we prove $X_0 \neq 0$ by contradiction. Assuming $X_0 = 0$ which means $v = v'i$, where $v'$ is a real vector. Then, $Av = Av'i = (a + bi)v'i = av'i - bv'$. Since $A$ is a real matrix, then vector $Av'i$ only consists of pure imaginary numbers, leading to $b = 0$. Then the contradiction appears since $\lambda' = a + bi$ is a complex number. According to previous analysis, one of $x_0$, $x_{-1}$, $y_{0,1} - y_{0,2}$ and $y_{-1,1} - y_{-1,2}$ is not 0. Here we analyze the case when $c = |y_{0,1} - y_{0,2}|$ is not equal to zero and omit other cases because these analyses are very similar. According to the iterative process, we have
$$X_{2t} = \tilde{\mathcal{A}}_{NM}^t X_0$$
$$= \tilde{\mathcal{A}}_{NM}^t (v + \bar{v})$$
$$= (a + bi)^t v + (a - bi)^t \bar{v}$$
$$= e^{it\theta}(a^2 + b^2)^{\frac{t}{2}} v + e^{-it\theta}(a^2 + b^2)^{\frac{t}{2}} \bar{v},$$
where $\theta = \text{sign}(b)\frac{\pi}{2}$ if $a = 0$, otherwise $\theta = \arctan(\frac{b}{a})$. Since $A_1 = [1, -1]$, then,
$$\|A_1 y_t\|_2 = |y_{t,1} - y_{t,2}|$$
$$= |2c(e^{it\theta} + e^{-it\theta})(a^2 + b^2)^{\frac{t}{2}}|$$
$$= |4c \cdot \cos(t\theta)|(a^2 + b^2)^{\frac{t}{2}}.$$

For $\cos(t\theta)$, either $\cos(t\theta) \equiv 1$ when $\theta = 0$, or $\lim_{t \to +\infty} \cos(t\theta)$ doesn't exist, which means there exists a constant $\delta > 0$ and $\{t_j\}_{j=1,2,\dots}$, where $\{t_j\}_{j=1,2,\dots}$ is a sequence that goes to infinity, such that $|\cos(t_j\theta)| > \delta$. We know that $|\lambda'| = (a^2 + b^2)^{\frac{1}{2}}$, then $|\lambda'| > 1$. Let $\lambda = |\lambda'|^{\frac{1}{2}}$, leading to $\lambda > 1$. In addition to $c \neq 0$, we have

$$\Delta_{1,t_j} \geq \|A_1 y_{t_j}\|_2 \geq \delta\lambda^t \in \Omega(\lambda^t).$$

Thus $\sup_{s \in [t]} \Delta_{1,s} \in \Omega(\lambda^t)$, and $\sup_{s \in [t]} \Delta_{2,s} \in \Omega(\lambda^t)$ can be proven in the same way. $\qquad \square$

### C.3 Optimistic Gradient Descent Ascent

In this subsection, we consider optimistic gradient descent ascent with step size $\eta$. Recall that the linear difference form of OGDA can be written as following:

$$
\begin{bmatrix} x_t \\ y_t \\ x_{t-1} \\ y_{t-1} \end{bmatrix} = \begin{bmatrix} I & -2\eta A_{t-1} & 0 & \eta A_{t-2} \\ 2\eta A_{t-1}^\top & I & -\eta A_{t-2}^\top & 0 \\ I & 0 & 0 & 0 \\ 0 & I & 0 & 0 \end{bmatrix} \begin{bmatrix} x_{t-1} \\ y_{t-1} \\ x_{t-2} \\ y_{t-2} \end{bmatrix}. \tag{18}
$$

We denote the matrix in (18) as $\mathcal{A}_t$ and let $X_t = (x_t^\top, y_t^\top, x_{t-1}^\top, y_{t-1}^\top)^\top$. Since payoff matrix has period of 2, by Floquet Theorem, we only have to analyze matrix $\mathcal{A}_{t+1}\mathcal{A}_t$. Let $\tilde{\mathcal{A}}_{OGDA} = \mathcal{A}_{t+1}\mathcal{A}_t$, then, we have

$$X_t = \tilde{\mathcal{A}}_{OGDA} X_{t-2}, \text{ for any } t \geq 2.$$

**Lemma C.2.** *For any step size $\eta > 0$, the spectral radius of $\tilde{\mathcal{A}}_{OGDA}$ is larger than $1$.*

*Proof.* Directly compute the characteristic polynomial $P_{\tilde{\mathcal{A}}_{OGDA}}(\lambda)$ of matrix $\tilde{\mathcal{A}}_{OGDA}$ gives

$$
\begin{aligned}
P_{\tilde{\mathcal{A}}_{OGDA}}(\lambda) &= \det(\lambda I - \tilde{\mathcal{A}}_{OGDA}) \\
&= \lambda \cdot (\lambda - 1) \cdot \left( \lambda - \left( 4\eta^2 - \frac{1}{2}\sqrt{64\eta^4 + 8\eta^2 + 1} + \frac{1}{2} \right) \right)^2 \\
&\quad \cdot \left( \lambda - \left( 4\eta^2 + \frac{1}{2}\sqrt{64\eta^4 + 8\eta^2 + 1} + \frac{1}{2} \right) \right)^2.
\end{aligned}
$$

Then, $\tilde{\mathcal{A}}_{OGDA}$ has an eigenvalue $\lambda' = 4\eta^2 + \frac{1}{2}\sqrt{64\eta^4 + 8\eta^2 + 1} + \frac{1}{2}$. It is easy to verify that $\lambda'$ is strictly monotonically increasing with $\eta \in [0, +\infty)$, and $\lambda'$ equals to $1$ iff $\eta = 0$. Since step size $\eta > 0$, thus the spectral radius of matrix $\tilde{\mathcal{A}}_{OGDA}$ is larger than $1$. $\qquad \square$

Now we are ready to prove Theorem 3.2 for the part of optimistic gradient descent ascent method.

*proof of Theorem 3.2.* **(part II, OGDA).** Let $X_0 = [x_0, y_{0,1}, y_{0,2}, x_{-1}, y_{-1,1}, y_{-1,2}]^\top$ be the eigenvector corresponding to the eigenvalue $\lambda'$ defined above. Then, it is directly to verify $x_0, x_{-1} \neq 0$.

In the iterative process, we have

$$X_{2t} = \tilde{\mathcal{A}}_{OGDA}^t X_0 = (\lambda')^t X_0, \ x_{2t} = (\lambda')^t x_0.$$

Since $A_1 = [1, -1]$, then

$$
\begin{aligned}
A_1^\top x_t &= [x_t, -x_t]^\top, A_1 y_t = y_{t,1} - y_{t,2}. \\
\Rightarrow \Delta_{1,t} &= \|A_1^\top x_t\|_2 + \|A_1 y_t\|_2 \\
&= \sqrt{2}|x_t| + |y_{t,1} - y_{t,2}| \\
&\geq \sqrt{2}\lambda'^{\frac{t}{2}} \min\{x_0, x_{-1}\}.
\end{aligned}
$$

By Theorem C.2, $\lambda' > 1$. Let $\lambda = (\lambda')^{\frac{1}{2}}$, then $\lambda > 1$. According to the inequality above, we have

$$\sup_{s \in [t]} \Delta_{1,s} \geq \Delta_{1,t} \geq \sqrt{2}\min\{x_0, x_{-1}\}\lambda^t \in \Omega\left(\lambda^t\right).$$

Similarly, we have $\sup_{s \in [t]} \Delta_{2,s} \in \Omega\left(\lambda^t\right)$. $\qquad \square$

## D Proof for convergent perturbed games with invertible payoff matrix

In this section, we provide a proof of a special case of Theorem 3.3 under the assumption that the payoff matrix is an invertible square matrix. Furthermore, we can demonstrate that this assumption leads to an exponential convergence rate.

**Proposition D.1.** *When the payoff matrix $A$ of the stable game is an invertible square matrix and $\lim_{t\to\infty} B_t = 0$, we have $\lim_{t\to\infty}(x_t, y_t) = (\mathbf{0}, \mathbf{0}) \in \mathbb{R}^{2n}$ in (OGDA), (EG), and (NM) with an exponential rate.*

*Proof.* According to Perron Theorem 2.5, we only need to prove maximum modulus of eigenvalues of iterative matrix $\mathcal{A}$ is less than 1. Lemma A.2 indicates that if the parameter condition on step sizes is satisfied, we have maximum modulus of eigenvalues of iterative matrix $\mathcal{A}$ is less than 1. This complete the proof. □

The above proof cannot be generalized to non-invertible matrices, as we have shown in Lemma A.2 that when the payoff matrix is non-invertible, then iterative matrices of the difference system associated with the game dynamics must have an eigenvalue equals to 1.

In the following, we prove Theorem 3.3 for the general case.

## E Omitted Proofs from Theorem 3.3

**Theorem 3.3.** *Assume that the (BAP assumption) holds, i.e., $\sum_{t=0}^{\infty} \|B_t\|_2$ is bounded, and let $\sigma$ be the maximum modulus of the singular value of payoff matrix $A$, then with parameters choice:*

- *for extra-gradient with step size $\alpha = \eta < \frac{1}{2\sigma}$,*

- *for optimistic gradient descent ascent with step size $\eta < \frac{1}{2\sigma}$,*

- *for negative momentum method with step size $\eta < \frac{1}{\sigma}$ and momentum parameters $\beta_1 = -\frac{1}{2}$ and $\beta_2 = 0$,*

*we have $\Delta_t$ converge to 0 with rate $\mathcal{O}(f(t))$. Here*

$$f(t) = \max\{\lambda^t, \sum_{i=t/2}^{\infty} \|B_i\|_2\},$$

*and $\lambda \in (0, 1)$ is determined by the eigenvalues of the iterative matrix of corresponding learning dynamics and the payoff matrix $A$ of the stable game.*

We separate the proof into several lemmas. Before going into details, we present a road map of the proof in Figure (7).

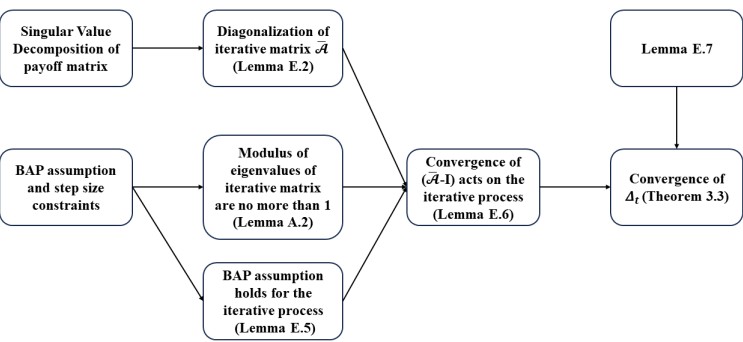

**Figure 7:** Road map for the prove of Theorem 3.3

As a first step, we demonstrate that the iterative matrices of learning dynamics can be diagonalized using singular value decomposition (SVD), as shown in Lemma E.2. This phenomenon was also shown in Gidel et al. [2019] for a general class of first order method. By singular value decomposition, we can write $A = U\Sigma_A V^\top$, where $U, V$ are unitary matrices, and $\Sigma_A$ is rectangular diagonal matrix with its diagonal entries being singular values of $A$. We denote this by

$$\Sigma_A = \begin{bmatrix} \sigma_{\mathbf{r}\times\mathbf{r}} & \mathbf{0}_{\mathbf{r}\times(\mathbf{m-r})} \\ \mathbf{0}_{(\mathbf{n-r})\times\mathbf{r}} & \mathbf{0}_{(\mathbf{n-r})\times(\mathbf{m-r})} \end{bmatrix} \in \mathbb{R}^{n\times m},$$

and

$$\sigma_{\mathbf{r}\times\mathbf{r}} = \begin{bmatrix} \sigma_1 & & \\ & \ddots & \\ & & \sigma_r \end{bmatrix} \in \mathbb{R}^{\mathbf{r}\times\mathbf{r}},$$

where $\sigma_i > 0$ are the singular values of $A$, $i \in [r]$. Let $\bar{x}_t = U^\top x_t$, $\bar{y}_t = V^\top y_t$, then we can transform the iterative process of three algorithms in convergent perturbed game into the equivalent form as followings :

**SVD formulation for OGDA in convergent perturbed game:**

$$\bar{x}_{t+1} = \bar{x}_t - 2\eta(\Sigma_A + U^\top B_t V)\bar{y}_t + \eta(\Sigma_A + U^\top B_{t-1}V)\bar{y}_{t-1},$$
$$\bar{y}_{t+1} = \bar{y}_t + 2\eta(\Sigma_A + V^\top B_t^\top U)\bar{x}_t - \eta(\Sigma_A + V^\top B_{t-1}^\top U)\bar{x}_{t-1}.$$

We represent the above in the form of a linear difference system:

$$\bar{X}_{t+1} = (\bar{\mathcal{A}} + \bar{\mathcal{B}}_t)\bar{X}_t, \tag{19}$$

where $\bar{X}_t = (\bar{x}_t^\top, \bar{y}_t^\top, \bar{x}_{t-1}^\top, \bar{y}_{t-1}^\top)^\top$,

$$\bar{\mathcal{A}} = \begin{bmatrix} I & -2\eta\Sigma_A & 0 & \eta\Sigma_A \\ 2\eta\Sigma_A^\top & I & -\eta\Sigma_A^\top & 0 \\ I & 0 & 0 & 0 \\ 0 & I & 0 & 0 \end{bmatrix} \in \mathbb{R}^{2(m+n)}, \tag{20}$$

and

$$\bar{\mathcal{B}}_t = \begin{bmatrix} 0 & -2\eta U^\top B_t V & 0 & \eta U^\top B_{t-1}V \\ 2\eta V^\top B_t^\top U & 0 & -\eta V^\top B_{t-1}^\top U & 0 \\ 0 & 0 & 0 & 0 \\ 0 & 0 & 0 & 0 \end{bmatrix} \in \mathbb{R}^{2(m+n)}. \tag{21}$$

**SVD formulation for EG in convergent perturbed game:**

$$\bar{x}_{t+1} = (I - \alpha\gamma(\Sigma_A\Sigma_A^\top + U^\top(AB_t^\top + B_tA^\top + B_tB_t^\top)U)\bar{x}_t - \alpha(\Sigma_A + U^\top B_t V)\bar{y}_t,$$
$$\bar{y}_{t+1} = (I - \alpha\gamma(\Sigma_A^\top\Sigma_A + V^\top(A^\top B_t + B_t^\top A + B_t^\top B_t)V)\bar{y}_t + \alpha(\Sigma_A^\top + V^\top B_t^\top U)\bar{x}_t.$$

We represent the above in the form of a linear difference system:

$$\bar{X}_{t+1} = (\bar{\mathcal{A}} + \bar{\mathcal{B}}_t)\bar{X}_t, \tag{22}$$

where $\bar{X}_t = (\bar{x}_t^\top, \bar{y}_t^\top)$,

$$\bar{\mathcal{A}} = \begin{bmatrix} I - \alpha\gamma\Sigma_A\Sigma_A^\top & -\alpha\Sigma_A \\ \alpha\Sigma_A^\top & I - \alpha\gamma\Sigma_A^\top\Sigma_A \end{bmatrix} \in \mathbb{R}^{m+n} \tag{23}$$

and
$$
\bar{\mathcal{B}}_t = \begin{bmatrix} -\alpha\gamma U^\top(AB_t^\top + B_tA^\top + B_tB_t^\top)U & -\alpha U^\top B_t V \\ \\ \alpha V^\top B_t^\top U & \alpha\gamma V^\top(A^\top B_t + B_t^\top A + B_t^\top B_t)V \end{bmatrix}.
\tag{24}
$$

**SVD formulation for NM in convergent perturbed game:**
$$
\begin{aligned}
\bar{x}_{t+1} =\ & (1+\beta_1)\bar{x}_t - \eta\left(\Sigma_A + U^\top B_t V\right)\bar{y}_t - \beta_1\bar{x}_{t-1}, \\
\bar{y}_{t+1} =\ & \left(I - \eta^2\left(\Sigma_A^\top\Sigma_A + V^\top(A^\top B_t + B_{t+1}^\top A + B_{t+1}^\top B_t)V\right)\right)\bar{y}_t \\
& + \eta\left(\Sigma_A^\top + V^\top B_{t+1}^\top U\right)((1+\beta_1)\bar{x}_t - \beta_1\bar{x}_{t-1}) - \beta_2\bar{y}_{t-1}.
\end{aligned}
$$

We represent the above in the form of a linear difference system:
$$
\bar{X}_{t+1} = (\bar{\mathcal{A}} + \bar{\mathcal{B}}_t)\bar{X}_t,
\tag{25}
$$
where $\bar{X}_t = (\bar{x}_t^\top, \bar{y}_t^\top, \bar{x}_{t-1}^\top, \bar{y}_{t-1}^\top)^\top$,
$$
\bar{\mathcal{A}} = \begin{bmatrix}
(1+\beta_1)I & -\eta\Sigma_A & -\beta_1 I & 0 \\
\eta(1+\beta_1)\Sigma_A^\top & I - \eta^2\Sigma_A^\top\Sigma_A & -\eta\beta_1\Sigma_A^\top & -\beta_2 I \\
I & 0 & 0 & 0 \\
0 & I & 0 & 0
\end{bmatrix} \in \mathbb{R}^{2(m+n)},
\tag{26}
$$
and
$$
\bar{\mathcal{B}}_t = \begin{bmatrix}
0 & -\eta U^\top B_t V & 0 & 0 \\
\eta(1+\beta_1)V^\top B_{t+1}^\top U & -\eta^2 V^\top(A^\top B_t + B_{t+1}^\top A + B_{t+1}^\top B_t)V & -\eta\beta_1 V^\top B_{t+1}^\top U & 0 \\
0 & 0 & 0 & 0 \\
0 & 0 & 0 & 0
\end{bmatrix}.
\tag{27}
$$

**Lemma E.1.** *The iterative matrix of SVD formulation for EG in convergent perturbed game in* (23) *is a normal matrix.*

*Proof.* Directly calculate shows
$$
\bar{\mathcal{A}}\bar{\mathcal{A}}^\top = \bar{\mathcal{A}}^\top\bar{\mathcal{A}} = \begin{bmatrix}
(I - \alpha\gamma\Sigma_A\Sigma_A^\top)^2 + \alpha^2\Sigma_A\Sigma_A^\top & 0 \\
0 & (I - \alpha\gamma\Sigma_A^\top\Sigma_A)^2 + \alpha^2\Sigma_A^\top\Sigma_A
\end{bmatrix}.
\tag{28}
$$
$\square$

**Lemma E.2.** *For a fixed payoff matrix A, the corresponding iterative matrices in* (20) *for OGDA,* (23) *for EG, and* (26) *for negative momentum method are diagonalizable.*

Note that the claim is true for EG since the iterative matrix is normal as we have shown in lemma E.1. Therefore, we will only consider the cases of OGDA and negative momentum method below. The idea behind proving these two claims is the same, we construct a set of linearly independent eigenvectors of (20) or (26) that form a basis of $\mathbb{R}^{2(m+n)}$, and under this basis, (20) or (26) can be represented by a diagonal matrix.

*Proof.* We firstly define some notation. In the following, we denote $e_i^n$ as an $n$-dimensional unit vector with 1 in the $i$-th position and 0 in other positions and denote $\sigma_p$ as the $p$-th singular value of the payoff matrix $A$ of the stable game, and denote $r$ as the rank of $A$. Thus for $p \in [r]$, $\sigma_p > 0$, and otherwise $\sigma_p = 0$. We will also denote the $n$-dimensional ($m$-dimensional) zero vector as $0^n (0^m)$.

**Part I, Diagonalization of (20):** Now we consider the diagonalization of matrix in (20). Recall that

$$\Sigma_A = \begin{bmatrix} \sigma_{\mathbf{r \times r}} & \mathbf{0}_{\mathbf{r \times (m-r)}} \\ \mathbf{0}_{\mathbf{(n-r) \times r}} & \mathbf{0}_{\mathbf{(n-r) \times (m-r)}} \end{bmatrix} \in \mathbb{R}^{n \times m},$$

and

$$\sigma_{\mathbf{r \times r}} = \begin{bmatrix} \sigma_1 & & \\ & \ddots & \\ & & \sigma_r \end{bmatrix} \in \mathbb{R}^{\mathbf{r \times r}}.$$

To prove $\bar{A}$ is diagonalizable, we only need to find $2(n+m)$ linearly independent eigenvectors of the matrix.

Then we can check the equations below

$$\begin{cases} \Sigma_A e_p^m = \sigma_p e_p^n, & \text{for } 1 \le p \le r, \\ \\ \Sigma_A e_j^m = 0^n, & \text{for } r+1 \le j \le m, \end{cases}$$

and

$$\begin{cases} \Sigma_A^\top e_p^n = \sigma_p e_p^m, & \text{for } 1 \le p \le r, \\ \\ \Sigma_A^\top e_i^n = 0^m, & \text{for } r+1 \le i \le n. \end{cases}$$

Now we respectively construct the eigenvectors corresponding to each eigenvalue, and prove these $2(n+m)$ vectors are linearly independent, forming a basis of $\mathbb{R}^{2(n+m)}$.

**Case 1** Eigenvectors correspond to eigenvalue 1 :
It can be verified that for $r+1 \le i \le n$

$$v_{1,i} = \begin{bmatrix} e_i^n \\ 0^m \\ 0^n \\ 0^m \end{bmatrix},$$

and for $r+1 \le j \le m$

$$w_{1,j} = \begin{bmatrix} 0^n \\ e_j^m \\ 0^n \\ 0^m \end{bmatrix}$$

are eigenvectors of $\bar{A}$ belonging to eigenvalue 1.

**Case 2** Eigenvectors correspond to eigenvalue 0 :
It can be verified that for $r+1 \le i \le n$,

$$v_{0,i} = \begin{bmatrix} 0^n \\ 0^m \\ e_i^n \\ 0^m \end{bmatrix}$$

and for $r + 1 \le j \le m$,

$$w_{0,j} = \begin{bmatrix} 0^n \\ 0^m \\ 0^n \\ e_j^m \end{bmatrix}$$

are eigenvectors of $\bar{\mathcal{A}}$ belonging to eigenvalue 0.

**Case 3** Other eigenvectors :

For $p = 1, \cdots, r$, consider the roots of the following polynomial :

$$\lambda^2(\lambda - 1)^2 + \eta^2 \sigma_p^2 (1 - 2\lambda)^2 = 0 \tag{29}$$

where $\sigma_p$ is the $p$-th diagonal element of $\Sigma_A$, and the solution of these polynomials are eigenvalues of $\bar{\mathcal{A}}$.

We first claim that except for finite choices of $\eta$, equation (29) has four different non-zero roots, denote them as $\lambda_{p,q}$, $q = 1, 2, 3, 4$. That is because a quartic polynomial equation has multiple roots if and only if its discriminant polynomial, a homogeneous polynomial with degree 6 on the coefficients of the quartic polynomial equation, equals to 0. Since a degree 6 polynomial has at most 6 roots, thus if $\eta$ is not a root of this discriminant polynomial, (29) will not have multiple roots. In the following, we will choose $\eta$ such that (29) has no multiple roots. According to Lemma A.2, the modulus of these eigenvalues are less than 1.

Let

$$\alpha_{p,q} = \frac{\eta \sigma_p (1 - 2\lambda_{p,q})}{\lambda_{p,q}^2 - \lambda_{p,q}}.$$

It can be verified that for $1 \le p \le r$ and $q = 1, 2, 3, 4$,

$$u_{p,q} = \begin{bmatrix} \lambda_{p,q} \alpha_{p,q} e_p^n \\ \lambda_{p,q} e_p^m \\ \alpha_{p,q} e_p^n \\ e_p^m \end{bmatrix}$$

are the eigenvectors of $\bar{\mathcal{A}}$ corresponding to eigenvalue $\lambda_{p,q}$.

Then we have constructed $2(n + m)$ eigenvectors, now we prove they are linearly independent. Suppose there exists coefficients $k_{1,i}, k_{0,i}$, where $i = r+1, \cdots, n$, $g_{1,j}, g_{0,j}$, where $j = r+1, \cdots, m$ and $f_{p,q}$, where $p = 1, \cdots, r$ and $q = 1, 2, 3, 4$, such that

$$\sum_{i=r+1}^n k_{1,i} v_{1,i} + \sum_{j=r+1}^m g_{1,j} w_{1,j} + \sum_{i=r+1}^n k_{0,i} v_{0,i} + \sum_{j=r+1}^m g_{0,j} w_{0,j} + \sum_{p=1}^r \sum_{q=1}^4 f_{p,q} u_{p,q} = 0. \tag{30}$$

For $r + 1 \le i \le n$, only $v_{1,i}$ has non-zero element at the $i$-th position of vector, so $k_{1,i} = 0$.

For $r + 1 \le j \le m$, only $w_{1,j}$ has non-zero element at the $(j + n)$-th position of vector, so $g_{1,j} = 0$.

For $r + 1 \le i \le n$, only $v_{0,i}$ has non-zero element at the $(i + n + m)$-th position of vector, so $k_{0,i} = 0$.

For $r + 1 \le j \le m$, only $w_{0,j}$ has non-zero element at the $(j + 2n + m)$-th position of vector, so $g_{0,j} = 0$.

For $1 \le p \le r$, at the $p$-th position of vector, only $u_{p,q}$, where $q = 1, 2, 3, 4$ has non-zero element. So we can yield

$$\sum_{q=1}^4 f_{p,q} u_{p,q} = 0.$$

The above equation holds for $p = 1, \cdots, r$. Because the eigenvectors of different eigenvalues are linearly independent, we have $f_{p,q} = 0$, where $q = 1, 2, 3, 4$ and $p = 1, \cdots, r$. Now we have concluded that all coefficients in (30) are zero, thus these eigenvectors are linearly independent.

Let $P$ be the matrix whose columns are consisted by the eigenvectors of $\bar{\mathcal{A}}$ constructed above, and $D$ be the diagonal matrix whose diagonal elements are eigenvalues of $\bar{\mathcal{A}}$. After an appropriate order arrangement of columns on $P$ and elements on $D$, we have

$$\bar{\mathcal{A}}P = PD.$$

Moreover, as we have shown above, the columns of $P$ are linearly independent, therefore $P$ is invertible, which implies $\bar{\mathcal{A}}$ is diagonalizable.

**Part II, Diagonalization of** (26)**:** Now we consider the diagonalization of the matrix in (26) and denote it as $\bar{\mathcal{A}}$. Similiarly, to prove this matrix is diagonalizable, we only need to find $2(n+m)$ linearly independent eigenvectors of the matrix. Now we respectively construct the eigenvectors corresponding to each eigenvalue, and prove these $2(n+m)$ vectors are linearly independent, forming a basis of $\mathbb{R}^{2(n+m)}$.

**Case 1:** Eigenvectors correspond to eigenvalue 1 :
It can be verified that for $r + 1 \leq i \leq n$,

$$v_{1,i} = \begin{bmatrix} e_i^n \\ 0^m \\ e_i^n \\ 0^m \end{bmatrix}$$

and for $r + 1 \leq j \leq m$,

$$w_{1,j} = \begin{bmatrix} 0^n \\ e_j^m \\ 0^n \\ e_j^m \end{bmatrix}$$

are eigenvectors of $\bar{\mathcal{A}}$ belonging to eigenvalue 1.

**Case 2:** Eigenvectors correspond to eigenvalue $\beta_1$.
It can be verified that for $r + 1 \leq i \leq n$,

$$v_{\beta_1,i} = \begin{bmatrix} \beta_1 e_i^n \\ 0^m \\ e_i^n \\ 0^m \end{bmatrix}$$

are eigenvectors of $\bar{\mathcal{A}}$ corresponding to eigenvalue $\beta_1$.

**Case 3:** Eigenvectors correspond to eigenvalue $\beta_2$.
It can be verified that for $r + 1 \leq j \leq m$,

$$w_{\beta_2,j} = \begin{bmatrix} 0^n \\ \beta_2 e_j^m \\ 0^n \\ e_j^m \end{bmatrix}$$

are eigenvectors of $\bar{\mathcal{A}}$ corresponding to eigenvalue $\beta_2$.

**Case 4:** Other eigenvectors.

For $p = 1, \cdots, r$, consider four roots of polynomial

$$(\lambda - 1)^2(\lambda - \beta_1)(\lambda - \beta_2) + \eta^2 \sigma_p^2 \lambda^3 = 0 \tag{31}$$

where $\sigma_p$ is the $p$-th diagonal element of $\Sigma_A$.

Now we consider the effect of different value of $\beta_1$ and $\beta_2$. If $\beta_1 = 0$ and $\beta_2 = 0$, the model degenerates to gradient descent algorithm, we only consider when $\beta_1$ and $\beta_2$ are not both zero. Similar to the situation in the Case 3 of diagonalization of (20), except for several values for $\eta$, equation (31) has four different roots, denote them as $\lambda_{p,q}$, $q = 1, 2, 3, 4$. If $\lambda_{p,q} \neq 0$, for $q = 1, 2, 3, 4$, let

$$\alpha_{p,q} = \frac{-\lambda_{p,q}^2 + (1 + \beta_1)\lambda_{p,q} - \beta_1}{\eta \sigma_p \lambda_{p,q}}.$$

We can check that

$$u_{p,q} = \begin{bmatrix} \lambda_{p,q} e_p^n \\ \lambda_{p,q}\alpha_{p,q} e_p^m \\ e_p^n \\ \alpha_{p,q} e_p^m \end{bmatrix}$$

is the eigenvector of $\bar{A}$ corresponding to eigenvalue $\lambda_{p,q}$, that is $\bar{A}u_{p,q} = \lambda_{p,q}u_{p,q}$. Else if $\lambda_{p,q} = 0$, that means either $\beta_1 = 0$ or $\beta_2 = 0$.

If $\beta_1 = 0$,

$$u_{p,q} = \begin{bmatrix} 0^n \\ 0^m \\ e_p^n \\ 0^m \end{bmatrix}$$

is the eigenvector of $\bar{A}$ corresponding to eigenvalue 0.

If $\beta_2 = 0$,

$$u_{p,q} = \begin{bmatrix} 0^n \\ 0^m \\ 0^n \\ e_p^m \end{bmatrix}$$

is the eigenvector of $\bar{A}$ corresponding to eigenvalue 0.

Now we obtain $2(n + m)$ eigenvectors, in the following we will prove these $2(n + m)$ eigenvectors are linearly independent. Suppose there exists coefficients $k_{1,i}, k_{\beta_1,i}$, where $i = r + 1, \cdots, n$, $g_{1,j}, g_{\beta_2,j}$, where $j = r + 1, \cdots, m$ and $f_{p,q}$, where $p = 1, \cdots, r$ and $q = 1, 2, 3, 4$, such that

$$\sum_{i=r+1}^{n} k_{1,i}v_{1,i} + \sum_{j=r+1}^{m} g_{1,j}w_{1,j} + \sum_{i=r+1}^{n} k_{\beta_1,i}v_{\beta_1,i} + \sum_{j=r+1}^{m} g_{\beta_2,j}v_{\beta_2,j} + \sum_{p=1}^{r}\sum_{q=1}^{4} f_{p,q}u_{p,q} = 0. \tag{32}$$

First we prove $f_{p,q} = 0$ for $p = 1, \cdots, r$ and $q = 1, 2, 3, 4$. If $\beta_1 = 0$, let $l = 1, \cdots, r$, then at the $l + n + m$-th position of vector, only $u_{l,q}$, $q = 1, 2, 3, 4$ has non-zero element. Else if $\beta_1 \neq 0$, let

$l = 1, \cdots, r$, then at the $l + 2n + m$-th position of vector, only $u_{l,q}$, $q = 1, 2, 3, 4$ has non-zero element. For these two case we both have

$$\sum_{q=1}^{4} f_{p,q} u_{p,q} = 0.$$

The above equation holds for $p = 1, \cdots, r$. Because the eigenvectors of different eigenvalues are linearly independent, we have $f_{p,q} = 0$, where $q = 1, 2, 3, 4$ and $p = 1, \cdots, r$.

For $i = r + 1, \cdots, n$, at the $i$-th position of vector, only $v_{1,i}$ and $v_{\beta_1,i}$ has non-zero element at this position, so we obtain

$$k_{1,i} v_{1,i} + k_{\beta_1,i} v_{\beta_1,i} = 0.$$

Notice that $\beta_1 \neq 1$, this means $k_{1,i} = 0$ and $k_{\beta_1,i} = 0$, where $i = r + 1, \cdots, n$.

For $j = r + 1, \cdots, m$, at the $j$-th position of vector, only $w_{1,j}$ and $w_{\beta_2,j}$ has non-zero element, so we can yield

$$g_{1,j} w_{1,j} + g_{\beta_2,j} w_{\beta_2,j} = 0.$$

Similarly, because of $\beta_2 \neq 1$, this means $g_{1,j} = 0$ and $g_{\beta_2,j} = 0$, where $j = r + 1, \cdots, m$.

We prove that if (32) holds, then all coefficients are zero, which illustrates that these eigenvectors are linearly independent. Same as the argument in Part $I$ of the proof, the existence of these $2(m + n)$ eigenvectors implies $\bar{\mathcal{A}}$ is diagonalizable.

$\square$

**Remark E.3.** *For payoff matrix $A$, given its SVD decomposition $A = U\Sigma_A V^\top$, let*

$$Q = \begin{bmatrix} U & & & \\ & V & & \\ & & U & \\ & & & V \end{bmatrix},$$

*then $Q$ is a unitary matrix. Furthermore, it can be verified that*

$$Q^\top \mathcal{A} Q = \bar{\mathcal{A}}$$

*for both OGDA and negative momentum method. That means*

1. *$\bar{\mathcal{A}}$ in (20) is diagonalizable implies that $\mathcal{A}$ in (8) is diagonalizable.*

2. *$\bar{\mathcal{A}}$ in (26) is diagonalizable implies that $\mathcal{A}$ in (11) is diagonalizable.*

**Lemma E.4** (Gronwall inequality, Colonius and Kliemann [2014]). *Let for all $t \in \mathbb{N}$, the functions $u, p, q, f : \mathbb{N} \to \mathbb{R}$ satisfy*

$$u(t) \leq p(t) + q(t) \sum_{\ell=a}^{t-1} f(\ell) u(\ell).$$

*Then, for all $t \in \mathbb{N}$*

$$u(t) \leq p(t) + q(t) \sum_{\ell=a}^{t-1} p(\ell) f(\ell) \prod_{\tau=\ell+1}^{k-1} (1 + q(\tau) f(\tau)). \qquad \text{(Gronwall inequality)}$$

Gronwall inequality is a useful tool to treat linear difference equations, it also has an analogy in continuous time case. For more about Gronwall inequality, see Lemma 6.1.3 in Colonius and Kliemann [2014].

**Lemma E.5.** *If $\{B_t\}_t$ satisfy the BAP assumption, i.e., $\sum_{t=1}^{\infty} \|B_t\|_2$ is bounded, then $\{\bar{\mathcal{B}}_t\}_t$ defined in (21), (24) and (27) also satisfy BAP assumption.*

*Proof.* We claim there exists some constant $c$, such that for any $t$, $\|\bar{\mathcal{B}}_{t-1}\|_2 \leq c \left( \|B_t\|_2 + \|B_t\|_2 + \|B_{t+1}\|_2 \right)$. With this property, we have

$$\sum_{t=0}^{\infty} \|\bar{\mathcal{B}}_t\|_2 \leq 3c \sum_{t=0}^{\infty} \|B_t\|_2 < +\infty,$$

then we prove the statement. In the following, we prove above claim for OGDA, EG, and negative momentum method.

**Case of OGDA :** We consider the matrix (21)

$$
\bar{\mathcal{B}}_t = \begin{bmatrix} 0 & -2\eta U^\top B_t V & 0 & \eta U^\top B_{t-1} V \\ 2\eta V^\top B_t^\top U & 0 & -\eta V^\top B_{t-1}^\top U & 0 \\ 0 & 0 & 0 & 0 \\ 0 & 0 & 0 & 0 \end{bmatrix}
$$

$$
= 2\eta \begin{bmatrix} 0 & -U^\top B_t V & 0 & 0 \\ V^\top B_t^\top U & 0 & 0 & 0 \\ 0 & 0 & 0 & 0 \\ 0 & 0 & 0 & 0 \end{bmatrix} + \eta \begin{bmatrix} 0 & 0 & 0 & U^\top B_{t-1} V \\ 0 & 0 & -V^\top B_{t-1}^\top U & 0 \\ 0 & 0 & 0 & 0 \\ 0 & 0 & 0 & 0 \end{bmatrix}.
$$

Denote the first matrix in right side of the equation as $H_1$, and the second one as $H_2$. From the above equation, we can obtain that $\|\bar{\mathcal{B}}_t\|_2 \le \|H_1\|_2 + \|H_2\|_2$. Recall the definition of 2-norm of matrix,

$$
\|H_1\|_2 = \max \sqrt{\text{Eigenvalue}\{H_1^\top H_1\}},
$$

$$
\|H_2\|_2 = \max \sqrt{\text{Eigenvalue}\{H_2^\top H_2\}},
$$

then,

$$
H_1^\top H_1 = 4\eta^2 \begin{bmatrix} U^\top B_t B_t^\top U & 0 & 0 & 0 \\ 0 & V^\top B_t^\top B_t V & 0 & 0 \\ 0 & 0 & 0 & 0 \\ 0 & 0 & 0 & 0 \end{bmatrix}
$$

and

$$
H_2^\top H_2 = \eta^2 \begin{bmatrix} 0 & 0 & 0 & 0 \\ 0 & 0 & 0 & 0 \\ 0 & 0 & U^\top B_{t-1} B_{t-1}^\top U & 0 \\ 0 & 0 & 0 & V^\top B_{t-1}^\top B_{t-1} V \end{bmatrix}.
$$

Because $U$ and $V$ are unitary matrices, we have

$$
\|H_1\|_2 = \max \sqrt{\text{Eigenvalue}\{H_1^\top H_1\}} = 4\eta^2 \max \sqrt{\text{Eigenvalue}\{B_t^\top B_t\}} = 4\eta^2 \|B_t\|_2
$$

and

$$
\|H_2\|_2 = \max \sqrt{\text{Eigenvalue}\{H_2^\top H_2\}} = \eta^2 \max \sqrt{\text{Eigenvalue}\{B_{t-1}^\top B_{t-1}\}} = \eta^2 \|B_{t-1}\|_2.
$$

Let $c = 4\eta^2$, then

$$
\|\bar{\mathcal{B}}_t\|_2 \le c \cdot (\|B_t\|_2 + \|B_{t-1}\|_2),
$$

we have completed the proof for OGDA.

**Case of EG :**  We consider the matrix (24)

$$\bar{\mathcal{B}}_t = \begin{bmatrix} -\alpha\gamma U^\top(AB_t^\top + B_tA^\top + B_tB_t^\top)U & -\alpha U^\top B_t V \\ \alpha V^\top B_t^\top U & \alpha\gamma V^\top(A^\top B_t + B_t^\top A + B_t^\top B_t)V \end{bmatrix}$$

$$= \begin{bmatrix} -\alpha\gamma U^\top(AB_t^\top + B_tA^\top + B_tB_t^\top)U & 0 \\ 0 & 0 \end{bmatrix}$$

$$+ \begin{bmatrix} 0 & 0 \\ 0 & \alpha\gamma V^\top(A^\top B_t + B_t^\top A + B_t^\top B_t)V \end{bmatrix}$$

$$+ \begin{bmatrix} 0 & -\alpha U^\top B_t V \\ 0 & 0 \end{bmatrix} + \begin{bmatrix} 0 & 0 \\ \alpha V^\top B_t^\top U & 0 \end{bmatrix}.$$

We separate $\mathcal{B}_1$ into four matrices and denote these matrices in right side of the equation as $H_1$, $H_2$, $H_3$ and $H_4$, respectively. Then

$$\|\bar{\mathcal{B}}_t\|_2 \leq \|H_1\|_2 + \|H_2\|_2 + \|H_3\|_2 + \|H_4\|_2.$$

Since $\sum_{t=1}^{\infty}\|B_t\|_2 \leq c$, then $\|B_t\|_2 \leq c$ for any $t$. We also assume that $c_2 = \|A\|_2$.

Then we have,

$$\|H_1\|_2 = \alpha\gamma\|U^\top(AB_t^\top + B_tA^\top + B_tB_t^\top)U\|_2$$

$$= \alpha\gamma\left(\|AB_t^\top + B_tA^\top + B_tB_t^\top\|_2\right)$$

$$\leq \alpha\gamma\left(\|A\|_2\|B_t\|_2 + \|A\|_2\|B_t\|_2 + \|B_t\|_2\|B_t\|_2\right)$$

$$\leq \alpha\gamma(2c_2 + c)\|B_t\|_2,$$

where the second equality is due to $U$ is unitary matrix. Similarly, $\|H_2\|_2 \leq \alpha\gamma(2c_2 + c)\|B_t\|_2$. In addition, $\|H_3\|_2 = \|H_4\|_2 = \alpha\|B_t\|_2$.

Thus for any $t$, we have the inequality between $\|\bar{\mathcal{B}}_t\|_2$ and $\|B_t\|_2$:

$$\|\bar{\mathcal{B}}_t\|_2 \leq \|H_1\|_2 + \|H_2\|_2 + \|H_3\|_2 + \|H_4\|_2 \leq \alpha\left((4c_2 + c)\gamma + 2\right)\|B_t\|_2.$$

Let $c_1 = c(4c_2 + c)\gamma + 2c$, summing the above inequality over $t$, we have

$$\sum_{t=1}^{\infty}\|\bar{\mathcal{B}}_t\|_2 \leq \left((4c_2 + c)\gamma + 2\right)\sum_{t=1}^{\infty}\|B_t\|_2 \leq c\left((4c_2 + c)\gamma + 2\right) = c_1.$$

**Case of Negative Momentum Method :**   We consider the matrix (27) ,

$$\bar{\mathcal{B}}_t = \begin{bmatrix} 0 & -\eta U^\top B_t V & 0 & 0 \\ \eta(1+\beta_1)V^\top B_{t+1}^\top U & -\eta^2 V^\top(A^\top B_t + B_{t+1}^\top A + B_{t+1}^\top B_t)V & -\eta\beta_1 V^\top B_{t+1}^\top U & 0 \\ 0 & 0 & 0 & 0 \\ 0 & 0 & 0 & 0 \end{bmatrix}$$

$$= \begin{bmatrix} 0 & 0 & 0 & 0 \\ \eta(1+\beta_1)V^\top B_t^\top U & 0 & 0 & 0 \\ 0 & 0 & 0 & 0 \\ 0 & 0 & 0 & 0 \end{bmatrix} + \begin{bmatrix} 0 & -\eta U^\top B_t V & 0 & 0 \\ 0 & 0 & -\eta\beta_1 V^\top B_{t+1}^\top U & 0 \\ 0 & 0 & 0 & 0 \\ 0 & 0 & 0 & 0 \end{bmatrix}$$

$$+ \begin{bmatrix} 0 & 0 & 0 & 0 \\ 0 & -\eta^2 V^\top(A^\top B_t + B_{t+1}^\top A + B_{t+1}^\top B_t)V & 0 & 0 \\ 0 & 0 & 0 & 0 \\ 0 & 0 & 0 & 0 \end{bmatrix}.$$

Denote the first matrix at the right side of equation as $H_1$, the second one as $H_2$ and the third as $H_3$. Then we have $\|\bar{\mathcal{B}}_t\|_2 \leq \|H_1\|_2 + \|H_2\|_2 + \|H_3\|_2$.

By definition,

$$\|H_1\|_2 = \max \sqrt{\text{Eigenvalue}\{H_1^\top H_1\}},$$
$$\|H_2\|_2 = \max \sqrt{\text{Eigenvalue}\{H_2^\top H_2\}}.$$

Because $U$ and $V$ are unitary matrices,

$$H_1^\top H_1 = \eta^2(1+\beta_1)^2 \begin{bmatrix} U^\top B_t B_t^\top U & 0 & 0 & 0 \\ 0 & 0 & 0 & 0 \\ 0 & 0 & 0 & 0 \\ 0 & 0 & 0 & 0 \end{bmatrix}$$

$$\Rightarrow \max \sqrt{\text{Eigenvalue}\{H_1^\top H_1\}} = \eta^2(1+\beta_1)^2 \max \sqrt{\text{Eigenvalue}\{B_t^\top B_t\}}$$

$$\Rightarrow \|H_1\|_2 = \eta^2(1+\beta_1)^2 \|B_t\|_2$$

and

$$H_2^\top H_2 = \eta^2 \begin{bmatrix} 0 & 0 & 0 & 0 \\ 0 & V^\top B_t^\top B_t V & 0 & 0 \\ 0 & 0 & \beta_1^2 U^\top B_{t+1} B_{t+1}^\top U & 0 \\ 0 & 0 & 0 & 0 \end{bmatrix}$$

$$\Rightarrow \max \sqrt{\text{Eigenvalue}\{H_2^\top H_2\}} = \eta^2 (\max \sqrt{\text{Eigenvalue}\{B_t^\top B_t\}} + \beta_1^2 \max \sqrt{\text{Eigenvalue}\{B_{t+1}^\top B_{t+1}\}})$$

$$\Rightarrow \|H_2\|_2 = \eta^2 (\|B_t\|_2 + \beta_1^2 \|B_{t+1}\|_2)$$

and

$$\|H_3\|_2 \le \eta^2 \left( \|A^\top B_t\|_2 + \|B_t^\top A\|_2 + \|B_t^\top B_t\|_2 \right)$$

$$\le \eta^2 \left( \|A\|_2\|B_t\|_2 + \|A\|_2\|B_t\|_2 + \|B_t\|_2\|B_t\|_2 \right)$$

$$\le c\|B_t\|_2,$$

where $c = 2\|A\|_2 + c'$, $c' = \max_{t \ge 0}\|B_t\|_2$. From $\sum_{t=0}^\infty \|B_t\|_2 < +\infty$ and $\|B_t\|_2 \ge 0$, we know $c'$ is a bounded constant. By combining the bounds for $H_1$, $H_2$ and $H_3$, we have completed the proof for negative momentum method. $\square$

**Lemma E.6.** *Assume that there exists a constant $c$ such that $\sum_{t=1}^\infty \|\bar{\mathcal{B}}_t\|_2 \le c$, and $\bar{\mathcal{A}}$ is as defined in (20), (23), or (26), then $\|(\bar{\mathcal{A}} - I)\bar{X}_t\|_2$ converges to 0 with rate $\mathcal{O}(f(t))$, where*

$$f(t) = \max\{\lambda^t, \sum_{i=t/2}^\infty \|B_i\|_2\}.$$

*Here $\lambda \in (0,1)$ is determined by the eigenvalues of the iterative matrix $\bar{\mathcal{A}}$ of corresponding learning dynamics and the payoff matrix $A$ of the stable game.*

*Proof.* Recall that we denote the SVD formulation of iterative process in (19) (22) and (25) as follows:

$$\bar{X}_{t+1} = (\bar{\mathcal{A}} + \bar{\mathcal{B}}_t)\bar{X}_t,$$

Since $\bar{\mathcal{A}}$ is a diagonalizable matrix from Lemma E.2, thus, there exists an invertible matrix $P$ such that $P\bar{\mathcal{A}}P^{-1} = D$, where $D$ is a diagonal matrix with the eigenvalues of $\bar{\mathcal{A}}$ as its entries. Since maximum modulus of eigenvalues of iterative matrix $\bar{\mathcal{A}}$ is no more than 1, then $\|D\|_2 \le 1$. Let

$$\hat{X}_t = P\bar{X}_t, \text{ and } \hat{\mathcal{B}}_t = P\bar{\mathcal{B}}_t P^{-1},$$

then the iterative process becomes $\hat{X}_{t+1} = (D + \hat{\mathcal{B}}_t)\hat{X}_t$.

By induction, we have

$$\hat{X}_t = (D + \hat{\mathcal{B}}_{t-1})\hat{X}_{t-1} = D^t \hat{X}_0 + \sum_{l=1}^t D^{t-l}\hat{\mathcal{B}}_{l-1}\hat{X}_{l-1} \tag{33}$$

$$\Longrightarrow \hat{\mathcal{B}}_t \hat{X}_t = \hat{\mathcal{B}}_t D^t \hat{X}_0 + \hat{\mathcal{B}}_t \sum_{l=1}^t D^{t-l}\hat{\mathcal{B}}_{l-1}\hat{X}_{l-1}.$$

Since $\|D^l\|_2 \le \|D\|_2^l \le 1$ for any $l \in [t]$, taking norm on both sides, we have

$$\|\hat{\mathcal{B}}_t \hat{X}_t\|_2 \le \|\hat{\mathcal{B}}_t\|_2\|\hat{X}_0\|_2 + \|\hat{\mathcal{B}}_t\|_2 \sum_{l=1}^t \|\hat{\mathcal{B}}_{l-1}\hat{X}_{l-1}\|_2,$$

Now we apply Gronwall inequality, let $u_t = \|\hat{\mathcal{B}}_t \hat{X}_t\|_2$, $p_t = \|\hat{\mathcal{B}}_t\|_2\|\hat{X}_0\|_2$, $q_t = \|\hat{\mathcal{B}}_t\|_2$ and $f_t \equiv 1$ in Gronwall inequality, see Lemma E.4, then we have

$$\|\hat{\mathcal{B}}_t \hat{X}_t\|_2 \le \|\hat{\mathcal{B}}_t\|_2\|\hat{X}_0\|_2 + \|\hat{\mathcal{B}}_t\|_2 \left( \sum_{l=1}^t \|\hat{\mathcal{B}}_l\|_2 \prod_{k=l-1}^{t-l} (1 + \|\hat{\mathcal{B}}_l\|_2) \right)\|\hat{X}_0\|_2.$$

Let $c_1 = \|P\|_2\|P^{-1}\|_2$. According to the assumption, there exists a constant $c$ such that $\sum_{t=1}^\infty \|\bar{\mathcal{B}}_t\|_2 \le c$, then

$$\sum_{t=1}^\infty \|\hat{\mathcal{B}}_t\|_2 = \sum_{t=1}^\infty \|P\bar{\mathcal{B}}_t P^{-1}\|_2$$

$$\le \sum_{t=1}^\infty \|P\|_2\|\bar{\mathcal{B}}_t\|_2\|P^{-1}\|_2$$

$$\le c_1 c.$$

Note that $\hat{\mathcal{B}}_t = P\bar{\mathcal{B}}_t P^{-1}$, so $\|\hat{\mathcal{B}}_t\|_2 \le \|P\|_2 \|\bar{\mathcal{B}}_t\|_2 \|P^{-1}\|_2 \le c_1 \|\bar{\mathcal{B}}_t\|_2$. Since $e^x \ge 1 + x$ for $x \in \mathbb{R}$, we obtain

$$\prod_{t=1}^{\infty}(1 + \|\hat{\mathcal{B}}_t\|_2) \le \prod_{t=1}^{\infty} e^{\|\hat{\mathcal{B}}_t\|_2}$$
$$= e^{\sum_{t=1}^{\infty}\|\hat{\mathcal{B}}_t\|_2}$$
$$\le e^{c_1 c}.$$

Let $c_2 = (1 + c_1 c \cdot e^{c_1 c})\|\hat{X}_0\|_2$, then

$$\|\hat{\mathcal{B}}_t \hat{X}_t\|_2 \le \|\hat{\mathcal{B}}_t\|_2 \|\hat{X}_0\|_2 \left(1 + \sum_{l=1}^{t} \|\hat{\mathcal{B}}_l\|_2 \prod_{k=l-1}^{t-l}(1 + \|\hat{\mathcal{B}}_l\|_2)\right)$$

$$\le \|\hat{\mathcal{B}}_t\|_2 \|\hat{X}_0\|_2 \left(1 + e^{c_1 c} \sum_{l=1}^{t} \|\hat{\mathcal{B}}_l\|_2\right)$$

$$\le \|\hat{\mathcal{B}}_t\|_2 \|\hat{X}_0\|_2 (1 + c_1 c \cdot e^{c_1 c})$$

$$\le c_2 \|\hat{\mathcal{B}}_t\|_2.$$

Multiplying $(D - I)$ on the equality (33) of both sides, we have

$$(D - I)\hat{X}_t = (D - I)D^t \hat{X}_0 + \sum_{l=1}^{t}(D - I)D^{t-l}\hat{\mathcal{B}}_{l-1}\hat{X}_{l-1}.$$

Let $c_3 = \max\{c_2, c_2 \sum_{l=1}^{\frac{1}{2}t}\|\hat{\mathcal{B}}_l\|_2\}$, taking the norm on both sides, we have

$$\|(D - I)\hat{X}_t\|_2 \le \delta^t \|\hat{X}_0\|_2 + \sum_{l=1}^{t} \delta^{t-l}\|\hat{\mathcal{B}}_{l-1}\hat{X}_{l-1}\|_2 \le \delta^t \|\hat{X}_0\|_2 + \sum_{l=1}^{t} c_2 \delta^{t-l}\|\hat{\mathcal{B}}_l\|_2$$

$$\le \delta^{\frac{1}{2}t}\left(\|\hat{X}_0\|_2 + c_2 \sum_{l=1}^{\frac{1}{2}t}\|\hat{\mathcal{B}}_l\|_2\right) + c_2 \sum_{l=\frac{1}{2}t}^{t}\|\hat{\mathcal{B}}_l\|_2$$

$$\le c_3 f(t).$$

Let $\lambda = \delta^{\frac{1}{2}}$, recall that $f(t) = \max\{\lambda^t, \sum_{i=t/2}^{\infty}\|B_i\|_2\}$. The last inequality is due to Lemma E.5, we can see that there is constant $c_4$ such that $\sum_{i=t/2}^{\infty}\|\hat{\mathcal{B}}_i\|_2 \le c_4 \sum_{i=t/2}^{\infty}\|B_i\|_2$. Recall that $f(t) = \max\{\lambda^t, \sum_{i=t/2}^{\infty}\|B_i\|_2\}$. Then, there exists a constant $c_5$ such that

$$\|(\bar{A} - I)\bar{X}_t\|_2 \le \|P^{-1}\|_2 \|(D - I)\hat{X}_t\|_2 \le c_5 f(t).$$

$\square$

**Lemma E.7.** *If $\|(\bar{A} - I)\bar{X}_t\|_2$ converges to 0 with rate $\mathcal{O}(f(t))$ as $t$ tends to infinity, then for OGD, EG and negative momentum method, $\|A^{\top}x_t\|_2 + \|Ay_t\|_2$ converges to 0 with rate $\mathcal{O}(f(t))$ when $t$ tends to infinity.*

*Proof of Lemma E.7.* We break the proof into three parts. Recall that $A = U\Sigma_A V^{\top}$, and $h = \|\Sigma_A\|_2$.

Firstly we prove the lemma for OGDA.

**OGDA** Writing $(\bar{\mathcal{A}} - I)\bar{X}_t$ into matrix form:

$$
\begin{bmatrix}
0 & -2\eta\Sigma_A & 0 & \eta\Sigma_A \\
2\eta\Sigma_A^\top & 0 & -\eta\Sigma_A^\top & 0 \\
I & 0 & -I & 0 \\
0 & I & 0 & -I
\end{bmatrix}
\begin{bmatrix}
U^\top x_{t-1} \\
V^\top y_{t-1} \\
U^\top x_{t-2} \\
V^\top y_{t-2}
\end{bmatrix}
=
\begin{bmatrix}
-2\eta\Sigma_A V^\top y_{t-1} + \eta\Sigma_A V^\top y_{t-2} \\
2\eta\Sigma_A U^\top x_{t-1} - \eta\Sigma_A U^\top x_{t-2} \\
U^\top x_{t-1} - U^\top x_{t-2} \\
V^\top y_{t-1} - V^\top y_{t-2}
\end{bmatrix}
$$

Since there is a constant $c$ such that $\|(\bar{\mathcal{A}} - I)\bar{X}_t\|_2 \le cf(t)$, then

$$\|2\eta\Sigma_A U^\top x_{t-1} - \eta\Sigma_A U^\top x_{t-2}\|_2 \le cf(t),$$

$$\|U^\top x_{t-1} - U^\top x_{t-2}\|_2 \le cf(t).$$

Using these two inequalities to bound $\|A^\top x_t\|_2$, we have

$$\|\eta\Sigma_A U^\top x_{t-1}\|_2$$

$$= \|2\eta\Sigma_A U^\top x_{t-1} - \eta\Sigma_A U^\top x_{t-2} - \eta\Sigma_A(U^\top x_{t-1} - U^\top x_{t-2})\|_2$$

$$\le \|2\eta\Sigma_A U^\top x_{t-1} - \eta\Sigma_A U^\top x_{t-2}\|_2 + \|\eta\Sigma_A(U^\top x_{t-1} - U^\top x_{t-2})\|_2$$

$$\le cf(t) + \eta cf(t).$$

Since $A^\top x_t = V\Sigma_A U^\top x_t$, then

$$\|A^\top x_t\|_2 = \|V\Sigma_A U^\top x_t\|_2$$

$$\le \|V\|_2 \|\Sigma_A U^\top x_t\|_2$$

$$\le \frac{ch(1+\eta)f(t)}{\eta},$$

where the last inequality is due to $\|\Sigma_A\|_2 = h$ and $V, U$ are unitary matrices. Similarly, we can obtain $\|Ay_t\|_2 \le \frac{(1+\eta)chf(t)}{\eta}$.

Next, we prove the lemma for extra-gradient.

**EG** Writing $(\bar{\mathcal{A}} - I)\bar{X}_t$ into matrix form:

$$
\begin{bmatrix}
-\alpha\gamma\Sigma_A\Sigma_A^\top & -\alpha\Sigma_A \\
\alpha\Sigma_A^\top & -\alpha\gamma\Sigma_A^\top\Sigma_A
\end{bmatrix}
\begin{bmatrix}
U^\top x_{t-1} \\
V^\top y_{t-1}
\end{bmatrix}
=
\begin{bmatrix}
-\alpha\gamma\Sigma_A\Sigma_A^\top U^\top x_{t-1} - \alpha\Sigma_A V^\top y_{t-1} \\
\alpha\Sigma_A^\top U^\top x_{t-1} - \alpha\gamma\Sigma_A^\top\Sigma_A V^\top y_{t-1}.
\end{bmatrix}
$$

Since there is a constant $c$ such that $\|(\bar{\mathcal{A}} - I)\bar{X}_t\|_2 \le cf(t)$, then

$$\|-\gamma\Sigma_A\Sigma_A^\top U^\top x_{t-1} - \Sigma_A V^\top y_{t-1}\|_2 \le \frac{cf(t)}{\alpha},$$

$$\|\Sigma_A^\top U^\top x_{t-1} - \gamma\Sigma_A^\top\Sigma_A V^\top y_{t-1}\|_2 \le \frac{cf(t)}{\alpha}.$$

Using these two inequalities to bound $\|A^\top x_t\|_2$, we have

$$\|(\gamma^2 \Sigma_A^\top \Sigma_A + I)\Sigma_A^\top U^\top x_{t-1}\|_2$$

$$=\|\Sigma_A^\top U^\top x_{t-1} - \gamma\Sigma_A^\top \Sigma_A V^\top y_{t-1} - \gamma\Sigma_A^\top \left(-\gamma\Sigma_A\Sigma_A^\top U^\top x_{t-1} - \Sigma_A V^\top y_{t-1}\right)\|_2$$

$$\leq\|\Sigma_A^\top U^\top x_{t-1} - \gamma\Sigma_A^\top \Sigma_A V^\top y_{t-1}\|_2 + \gamma\|\Sigma_A^\top\|_2\|-\gamma\Sigma_A\Sigma_A^\top U^\top x_{t-1} - \Sigma_A V^\top y_{t-1}\|_2$$

$$\leq\frac{(1+\gamma h)cf(t)}{\alpha}.$$

Since matrix $\gamma^2\Sigma_A^\top\Sigma_A + I$ is invertible, then

$$\|\Sigma_A U^\top x_t\|_2 = \|(\gamma^2\Sigma_A^\top\Sigma_A + I)^{-1}(\gamma^2\Sigma_A^\top\Sigma_A + I)\Sigma_A^\top U^\top x_{t-1}\|_2$$

$$\leq\|(\gamma^2\Sigma_A^\top\Sigma_A + I)^{-1}\|_2\|(\gamma^2\Sigma_A^\top\Sigma_A + I)\Sigma_A^\top U^\top x_{t-1}\|_2$$

$$\leq\|(\gamma^2\Sigma_A^\top\Sigma_A + I)\Sigma_A^\top U^\top x_{t-1}\|_2$$

$$\leq\frac{(1+\gamma h)cf(t)}{\alpha},$$

where the last inequality is due to $\|(\gamma^2\Sigma_A^\top\Sigma_A + I)^{-1}\|_2 \leq 1$. Since $A^\top x_t = V\Sigma_A U^\top x_t$, then

$$\|A^\top x_t\|_2 = \|V\Sigma_A U^\top x_t\|_2 \leq \|V\|_2\|\Sigma_A U^\top x_t\|_2 \leq \frac{(1+\gamma h)cf(t)}{\alpha},$$

where the last inequality is due to $V$ is unitary matrices. Similarly, we can obtain

$$\|Ay_t\|_2 \leq \frac{(1+\gamma h)}{\alpha}f(t)c.$$

Finally, we prove the lemma for negative momentum method.

**Negative Momentum Method** Writing $(\bar{\mathcal{A}} - I)\bar{X}_t$ into matrix form:

$$\begin{bmatrix} \beta_1 I & -\eta\Sigma_A & -\beta_1 I & 0 \\ \eta(1+\beta_1)\Sigma_A^\top & -\eta^2\Sigma_A^\top\Sigma_A & -\eta\beta_1\Sigma_A^\top & -\beta_2 I \\ I & 0 & -I & 0 \\ 0 & I & 0 & -I \end{bmatrix}\begin{bmatrix} U^\top x_{t-1} \\ V^\top y_{t-1} \\ U^\top x_{t-2} \\ V^\top y_{t-2} \end{bmatrix}$$

$$=\begin{bmatrix} \beta_1 U^\top x_{t-1} - \eta\Sigma_A V^\top y_{t-1} - \beta_1 U^\top x_{t-2} \\ \eta(1+\beta_1)\Sigma_A^\top U^\top x_{t-1} - \eta^2\Sigma_A^\top\Sigma_A V^\top y_{t-1} - \eta\beta_1\Sigma_A^\top U^\top x_{t-2} - \beta_2 V^\top y_{t-2} \\ U^\top x_{t-1} - U^\top x_{t-2} \\ V^\top y_{t-1} - V^\top y_{t-2} \end{bmatrix}$$

Since there is a constant $c$ such that $\|(\bar{\mathcal{A}} - I)\bar{y}_t\|_2 \leq cf(t)$, then

$$\|\beta_1 U^\top x_{t-1} - \eta\Sigma_A V^\top y_{t-1} - \beta_1 U^\top x_{t-2}\|_2 \leq cf(t),$$

$$\|U^\top x_{t-1} - U^\top x_{t-2}\|_2 \leq cf(t).$$

Using these two inequalities to bound $\|Ay_t\|_2$, we have

$$\|\eta\Sigma_A V^\top y_{t-1}\|_2$$

$$=\|\beta_1\left(U^\top x_{t-1}-U^\top x_{t-2}\right)-\left(\beta_1 U^\top x_{t-1}-\eta\Sigma_A V^\top y_{t-1}-\beta_1 U^\top x_{t-2}\right)\|_2$$

$$\leq\|\beta_1\left(U^\top x_{t-1}-U^\top x_{t-2}\right)\|_2+\|\beta_1 U^\top x_{t-1}-\eta\Sigma_A V^\top y_{t-1}-\beta_1 U^\top x_{t-2}\|_2$$

$$\leq\beta_1 cf(t)+cf(t).$$

Since $Ay_t=U\Sigma_A V^\top y_t$, then

$$\|Ay_t\|_2=\|U\Sigma_A V^\top y_t\|_2\leq\|U^\top\|_2\|\Sigma_A V^\top y_t\|_2\leq\frac{c(1+\beta_1)f(t)}{\eta},$$

where the last inequality is due to $\|U\|_2=1$.

We also have

$$\|\eta(1+\beta_1)\Sigma_A^\top U^\top x_{t-1}-\eta^2\Sigma_A^\top\Sigma_A V^\top y_{t-1}-\eta\beta_1\Sigma_A^\top U^\top x_{t-2}-\beta_2 V^\top y_{t-2}\|_2\leq cf(t)$$

and

$$\|V^\top y_{t-1}-V^\top y_{t-2}\|_2\leq cf(t).$$

Then,

$$\|\eta\Sigma_A\Sigma_A^\top U^\top x_{t-1}\|_2$$

$$=\|\Sigma_A\cdot\left([\eta(1+\beta_1)\Sigma_A^\top U^\top x_{t-1}-\eta^2\Sigma_A^\top\Sigma_A V^\top y_{t-1}-\eta\beta_1\Sigma_A^\top U^\top x_{t-2}-\beta_2 V^\top y_{t-2}]\right.$$
$$\left.-\eta\Sigma_A^\top\left(\beta_1 U^\top x_{t-1}-\eta\Sigma_A V^\top y_{t-1}-\beta_1 U^\top x_{t-2}\right)+\beta_2\Sigma_A^\top V^\top y_{t-2})\|_2$$

$$\leq\|\Sigma_A\|_2 cf(t)+\|\Sigma_A\|_2 cf(t)+\frac{\beta_2(1+\beta_1)}{\eta}cf(t)$$

$$\leq\left(2h+\frac{\beta_2(1+\beta_1)}{\eta}\right)cf(t)$$

Since $\|\Sigma_A\|_2=h$, then $\|\Sigma_A^\top U^\top x_t\|_2\leq\frac{2\eta h+\beta_2(1+\beta_1)}{\eta^2 h}cf(t)$. Since $A^\top x_t=V\Sigma_A U^\top x_t$, then

$$\begin{aligned}\|A^\top x_t\|_2&=\|V\Sigma_A U^\top x_t\|_2\\&\leq\|V\|_2\|\Sigma_A U^\top x_t\|_2\\&\leq\frac{2\eta h+\beta_2(1+\beta_1)}{\eta^2 h}cf(t).\end{aligned}$$

$\square$

Now we are ready to prove Theorem 3.3.

*proof of Theorem 3.3.* According to Lemma E.5, assumptions of Lemma E.6 have been satisfied by the difference equations associated to our learning dynamics, thus we have $\|(\mathcal{A}-I)\bar{X}_t\|_2$ converges to 0 with rate $f(t)$. Moreover, by Lemma E.7, $\Delta_t$ converges to 0 with rate $f(t)$ in OGD, EG and negative momentum method. We complete the proof. $\square$

# F   Omitted Proofs from Theorem 3.4

**Theorem 3.4.** *In a convergent perturbed game, if two players use Extra-gradient, there holds* $\lim_{t\to\infty} \Delta_t = 0$ *with step size* $\alpha = \eta < \frac{1}{2\sigma}$ *where* $\sigma$ *is the maximum modulus of the singular value of payoff matrix* $A$.

Recall that Extra Gradient satisfies the linear difference equation (3), denote the iterative matrix in equation (3) with payoff matrix $A_t$ as $\mathcal{A}_t$. According to the convergence of payoff matrix, we have $\lim_{t\to\infty} A_t = A$. Let $B_t = A_t - A$, then we have $\lim_{t\to\infty} B_t = 0$. Denote $\mathcal{A}$ as the iterative matrix when payoff matrix is time invariant and equal to $A$. Let

$$\mathcal{B}_t = \mathcal{A}_t - \mathcal{A}.$$

To prove the theorem, we first establish several necessary lemmas.

**Lemma F.1.** *Given* $\lim_{t\to\infty} B_t = 0$, *we have* $\lim_{t\to\infty} \mathcal{B}_t = 0$.

*Proof.* Recall that

$$\mathcal{A}_t = \begin{bmatrix} I - \alpha\gamma A_t A_t^\top & -\alpha A_t \\ \alpha A_t^\top & I - \gamma\alpha A_t^\top A_t \end{bmatrix}$$

and

$$\mathcal{A} = \begin{bmatrix} I - \alpha\gamma A A^\top & -\alpha A \\ \alpha A^\top & I - \gamma\alpha A^\top A \end{bmatrix},$$

we can obtain that

$$\mathcal{B}_t = \begin{bmatrix} -\alpha\gamma(AB_t^\top + B_t A^\top + B_t B_t^\top) & -\alpha B_t \\ \alpha B_t^\top & \alpha\gamma(A^\top B_t + B_t^\top A + B_t^\top B_t) \end{bmatrix}$$

$$= \begin{bmatrix} -\alpha\gamma(AB_t^\top + B_t A^\top + B_t B_t^\top) & 0 \\ 0 & 0 \end{bmatrix} + \begin{bmatrix} 0 & 0 \\ 0 & \alpha\gamma(A^\top B_t + B_t^\top A + B_t^\top B_t) \end{bmatrix}$$

$$+ \begin{bmatrix} 0 & -\alpha B_t \\ 0 & 0 \end{bmatrix} + \begin{bmatrix} 0 & 0 \\ \alpha B_t^\top & 0 \end{bmatrix}.$$

We separate $\mathcal{B}_t$ into four matrices and denote these matrices in right side of the equation as $H_1$, $H_2$, $H_3$ and $H_4$, respectively. Then

$$\|\mathcal{B}_t\|_2 \le \|H_1\|_2 + \|H_2\|_2 + \|H_3\|_2 + \|H_4\|_2.$$

Since $\lim_{t\to\infty} B_t = 0$, then $\lim_{t\to\infty} \|B_t\|_2 = 0$, so we can yield that there exists $c$ such that $\|B_t\|_2 \le c$ for any t. We also assume that $c_1 = \|A\|_2$.

Then we have

$$\|H_1\|_2 = \alpha\gamma\|AB_t^\top + B_t A^\top + B_t B_t^\top\|_2$$

$$\le \alpha\gamma \left(\|A\|_2\|B_t\|_2 + \|A\|_2\|B_t\|_2 + \|B_t\|_2\|B_t\|_2\right)$$

$$\le \alpha\gamma(2c_1 + c)\|B_t\|_2,$$

Similarly, $\|H_2\|_2 \le \alpha\gamma(2c_1 + c)\|B_t\|_2$. In addition, $\|H_3\|_2 = \|H_4\|_2 = \alpha\|B_t\|_2$.

Then we can obtain that there exists a constant $c_2$, such that $\|\mathcal{B}_t\|_2 \le c_2\|B_t\|_2$, which implies that $\lim_{t\to\infty} \mathcal{B}_t = 0$. □

With the lemma above, we directly utilize $\lim_{t\to\infty} \mathcal{B}_t = 0$ in proving Theorem 3.4.

**Lemma F.2.** *Let $X_t = (x_t^\top, y_t^\top)^\top$, then there exists $t_0 > 0$, such that when $t > t_0$, $\|X_t\|_2$ is monotonically non-increasing. Moreover, $\exists c_0 \geq 0$, $\lim_{t\to\infty} \|X_t\|_2 = c_0$.*

*Proof.* First we prove that there exists $t_0$ such that when $t > t_0$, $\|\mathcal{A}_t\|_2 \leq 1$.

From the proof of Lemma A.2, we know that if $\alpha = \gamma \leq \frac{1}{\sigma_t}$, where $\sigma_t$ is the maximal singular value of payoff matrix $A_t$, then the discriminant in equation (10) is satisfied, i.e., $\|\mathcal{A}_t\|_2 \leq 1$. Here we choose $\alpha = \gamma \leq \frac{1}{2\sigma}$, where $\sigma$ is the maximal singular value of payoff matrix $A$. Because $A_t$ converges to $A$, we can conclude that there exists $t_0$, such that when $t \geq t_0$, $\sigma_t < 2\sigma$. This implies that $\alpha = \gamma \leq \frac{1}{2\sigma} < \frac{1}{\sigma_t}$, which means $\|\mathcal{A}_t\|_2 \leq 1$. Therefore we prove that here exists $t_0$ such that when $t > t_0$, $\|X_t\|_2 \leq \|X_{t-1}\|_2$. For any $t > t_0$, we have

$$\|X_t\|_2 = \|\mathcal{A}_{t-1} X_{t-1}\|_2$$
$$\leq \|\mathcal{A}_{t-1}\|_2 \|X_{t-1}\|_2$$
$$\leq \|X_{t-1}\|_2.$$

Therefore, we have that when $t > t_0$, $\|X_t\|_2$ is monotonically non-increasing.

From the fact that $\|X_t\|_2$ is monotonically non-increasing and no smaller than 0, we obtain $\exists c_0 \geq 0$, $\lim_{t\to\infty} \|X_t\|_2 = c_0$. $\qquad\square$

In fact, the property that $\|X_t\|_2$ is monotonically non-increasing is closely related to the iterative matrix of EG is normal, which causes part of the difference between EG and OGDA or negative momentum method.

**Lemma F.3.** *Decompose $\mathbb{R}^{n+m} = V_1 \oplus V_2$ where $V_1$ is the eigenspace of eigenvalue $1$ of matrix $\mathcal{A}$, $V_1$ and $V_2$ are mutually perpendicular. Define $\lambda = \max_{s\neq 1, s\in Eigenvalue \mathcal{A}} |s|$. Then if $v \in V_2$, $\|\mathcal{A}v\|_2 \leq \lambda \|v\|_2$.*

*Proof.* Let $W_s = \{v \in \mathbb{R}^{n+m} \mid \mathcal{A}v = sv\}$, that is $W_s$ is the eigenspace of eigenvalue $s$ of $\mathcal{A}$. Let $V_1 = W_1$ and $V_2 = \oplus_{s\neq 1} W_s$. By $\mathcal{A}$ is normal, we have $\mathbb{R}^{n+m} = V_1 \oplus V_2$. $V_1$ and $V_2$ are mutually perpendicular.

Then we only need to prove if $v \in V_2$, $\|\mathcal{A}v\|_2 \leq \lambda \|v\|_2$. From Lemma A.2, we know that $\lambda < 1$. Because

$$V_2 = \oplus_{s\neq 1, s\in Eigenvalue \mathcal{A}} W_s,$$

$v$ can be decomposed as $v = \sum_{s\neq 1, s\in Eigenvalue \mathcal{A}} k_s w_s$, where $w_s \in W_s$, $k_s$ is the coefficient and for different $s_1$ and $s_2$ which are eigenvalues of $\mathcal{A}$, $w_{s_1}$ and $w_{s_2}$ are perpendicular.

Therefore $\|v\|_2^2 = \sum_{s\neq 1, s\in Eigenvalue \mathcal{A}} k_s^2 \|w_s\|_2^2$. Then $\mathcal{A}v = \sum_{s\neq 1, s\in Eigenvalue \mathcal{A}} k_s \mathcal{A}w_s$, therefore we have

$$\|\mathcal{A}v\|_2^2 = \sum_{s\neq 1, s\in Eigenvalue \mathcal{A}} k_s^2 \|\mathcal{A}w_s\|_2^2$$
$$= \sum_{s\neq 1, s\in Eigenvalue \mathcal{A}} |s|^2 k_s^2 \|w_s\|_2^2$$
$$\leq \lambda^2 \sum_{s\neq 1, s\in Eigenvalue \mathcal{A}} k_s^2 \|w_s\|_2^2$$
$$= \lambda^2 \|v\|_2^2$$

which means that $\|\mathcal{A}v\|_2 \leq \lambda \|v\|_2$, this complete the proof. $\qquad\square$

Now we can decompose $X_t = v_t^1 + v_t^2$ where $v_t^1 \in V_1$ and $v_t^2 \in V_2$. Similarly, we also decompose $\mathcal{B}_t X_t = w_t^1 + w_t^2$ where $w_t^1 \in V_1$ and $w_t^2 \in V_2$.

**Lemma F.4.** *If $\lim_{t\to\infty} \|v_t^2\|_2 = 0$, then $\lim_{t\to\infty} (A^\top x_t, Ay_t) = (\mathbf{0}, \mathbf{0})$, which implies that $\lim_{t\to\infty} \Delta_t = 0$.*

*Proof.* Let $v_t^1 = \begin{pmatrix} x_t^1 \\ y_t^1 \end{pmatrix}$, $v_t^2 = \begin{pmatrix} x_t^2 \\ y_t^2 \end{pmatrix}$, then by $X_t = \begin{pmatrix} x_t \\ y_t \end{pmatrix} = v_t^1 + v_t^2$, we have $x_t = x_t^1 + x_t^2$ and $y_t = y_t^1 + y_t^2$. First, we prove $A^\top x_t^1 = 0$ and $Ay_t^1 = 0$. By $v_t^1 \in V_1$, we have

$$
\begin{bmatrix} I - \alpha\gamma AA^\top & -\alpha A \\ \alpha A^\top & I - \alpha\gamma A^\top A \end{bmatrix} \begin{bmatrix} x_t^1 \\ y_t^1 \end{bmatrix} = \begin{bmatrix} x_t^1 \\ y_t^1 \end{bmatrix}
$$

that is

$$
\begin{bmatrix} x_t^1 - \alpha\gamma AA^\top x_t^1 - \alpha Ay_t^1 \\ \alpha A^\top x_t^1 + y_t^1 - \alpha\gamma A^\top Ay_t^1 \end{bmatrix} = \begin{bmatrix} x_t^1 \\ y_t^1 \end{bmatrix}
$$

$$
\implies \begin{cases} -\gamma AA^\top x_t^1 - Ay_t^1 = 0, \\ A^\top x_t^1 - \gamma A^\top Ay_t^1 = 0, \end{cases}
$$

$$
\implies \begin{cases} A^\top x_t^1 = 0, \\ Ay_t^1 = 0, \end{cases}
$$

where the second double arrow symbols is due to $\gamma^2 AA^\top + I$ is invertible.

According to $A^\top x_t^1 = 0$ and $Ay_t^1 = 0$, we have

$$
\begin{aligned}
\begin{bmatrix} A^\top x_t \\ Ay_t \end{bmatrix} = \begin{bmatrix} A^\top & \\ & A \end{bmatrix} X_t &= \begin{bmatrix} A^\top & \\ & A \end{bmatrix} (v_t^1 + v_t^2) \\
&= \begin{bmatrix} A^\top & \\ & A \end{bmatrix} \begin{bmatrix} x_t^1 \\ y_t^1 \end{bmatrix} + \begin{bmatrix} A^\top & \\ & A \end{bmatrix} v_t^2 \\
&= \begin{bmatrix} A^\top x_t^1 \\ Ay_t^1 \end{bmatrix} + \begin{bmatrix} A^\top & \\ & A \end{bmatrix} v_t^2 \\
&= \begin{bmatrix} A^\top & \\ & A \end{bmatrix} v_t^2.
\end{aligned}
$$

We can see that if $\lim_{t\to\infty} \|v_t^2\|_2 = 0$, then $\lim_{t\to\infty}(A^\top x_t, Ay_t) = (0,0)$. $\qquad\square$

Now we are ready to prove Theorem 3.4.

*proof of Theorem 3.4.* According to Lemma F.4, we directly obtain $\lim_{t\to\infty}\Delta_t = 0$ if $\lim_{t\to\infty}\|v_t^2\|_2 = 0$. In the following We prove $\lim_{t\to\infty}\|v_t^2\|_2 = 0$ by contradiction. Assuming that $\{\|v_t^2\|_2\}_t$ doesn't converge to 0, i.e.,

$$
\exists\delta > 0,\ \exists t_1, t_2, \cdots,\ s.t.\ \|v_{t_i}^2\|_2 > \delta \tag{34}
$$

where $t_i$ tends to $+\infty$ as $i \to \infty$.

Let $\epsilon = \frac{1}{8}\delta(1 - \lambda^2)$, then we can find such $t_j$ that for any $t > t_j$, $\|X_t\|_2^2 - \|X_{t+1}\|_2^2 \leq \epsilon$ by Lemma F.2 , latter we will prove that under the assumption (34), there exists $t > t_j$ such that $\|X_t\|_2^2 - \|X_{t+1}\|_2^2 > \epsilon$ which contradicts to Lemma F.2. Then we can also find a $t_k \geq t_j$ such that for any $t > t_k$,

$$
\|\mathcal{B}_t\|_2 \leq \min\{\frac{\delta(1 - \lambda^2)}{8\|X_0\|_2}, \frac{\delta(1 - \lambda^2)}{8\|X_0\|_2^2}\}
$$

by $\lim_{t\to\infty}\mathcal{B}_t = 0$, and for any $t_s \geq t_k$, $\|v_{t_s}^2\|_2 > \delta$. We choose such a $t_s$ and denote it as $t$.

Now we give the bound for $\|\mathcal{B}_t X_t\|_2$, $\|w_t^1\|_2$ and $\|w_t^2\|_2$,

$$\|\mathcal{B}_t X_t\|_2 \leq \|\mathcal{B}_t\|_2 \cdot \|X_t\|_2$$

$$\leq \min\{\frac{\delta(1-\lambda^2)}{8\|X_0\|_2}, \frac{\delta(1-\lambda^2)}{8\|X_0\|_2^2}\} \cdot \|X_0\|_2$$

$$= \min\{\frac{\delta(1-\lambda^2)}{8}, \frac{\delta(1-\lambda^2)}{8\|X_0\|_2}\}$$

$$\leq \frac{\delta(1-\lambda^2)}{8},$$

where the second inequality comes from Lemma F.2 . Together with $\|w_t^1\|_2$ and $\|w_t^2\|_2$ are perpendicular, which implies $\|\mathcal{B}_t X_t\|_2^2 = \|w_t^1\|_2^2 + \|w_t^2\|_2^2$, for $i = 1, 2$, we have

$$\|w_t^i\|_2 \leq \|\mathcal{B}_t X_t\|_2 \leq \frac{\delta(1-\lambda^2)}{8\|X_0\|_2}.$$

Now we try to determine the relationship between $X_{t+1}$ and $X_t$,

$$X_{t+1} = (\mathcal{A} + \mathcal{B}_t)X_t$$

$$= \mathcal{A}X_t + \mathcal{B}_t X_t$$

$$= \mathcal{A}(v_t^1 + v_t^2) + w_t^1 + w_t^2$$

$$= (v_t^1 + w_t^1) + (\mathcal{A}v_t^2 + w_t^2),$$

where $v_t^1 + w_t^1 \in V_1$ and $\mathcal{A}v_t^2 + w_t^2 \in V_2$, so $v_t^1 + w_t^1$ and $\mathcal{A}v_t^2 + w_t^2$ are perpendicular, and

$$\|X_{t+1}\|_2^2 = \|v_t^1 + w_t^1\|_2^2 + \|\mathcal{A}v_t^2 + w_t^2\|_2^2$$

$$\leq \|v_t^1\|_2^2 + \|w_t^1\|_2^2 + 2\|v_t^1\|_2\|w_t^1\|_2 + \|\mathcal{A}v_t^2\|_2^2 + \|w_t^2\|_2^2 + 2\|\mathcal{A}v_t^2\|_2\|w_t^2\|_2$$

$$\leq \|v_t^1\|_2^2 + \lambda^2\|v_t^2\|_2^2 + \|w_t^1\|_2^2 + \|w_t^2\|_2^2 + 2\|v_t^1\|_2\|w_t^1\|_2 + 2\lambda\|v_t^2\|_2\|w_t^2\|_2$$

$$\leq \|v_t^1\|_2^2 + \lambda^2\|v_t^2\|_2^2 + \|\mathcal{B}_t X_t\|_2^2 + 2\|X_0\|_2\frac{\delta(1-\lambda^2)}{8\|X_0\|_2} + 2\lambda\|X_0\|_2\frac{\delta(1-\lambda^2)}{8\|X_0\|_2}$$

$$\leq \|v_t^1\|_2^2 + \lambda^2\|v_t^2\|_2^2 + \frac{\delta(1-\lambda^2)}{8} + 2\frac{\delta(1-\lambda^2)}{8} + 2\lambda\frac{\delta(1-\lambda^2)}{8}$$

$$\leq \|v_t^1\|_2^2 + \lambda^2\|v_t^2\|_2^2 + \frac{5}{8}\delta(1-\lambda^2).$$

The second inequality comes from Lemma F.3, while the third inequality comes from $\|\mathcal{B}_t X_t\|_2^2 = \|w_t^1\|_2^2 + \|w_t^2\|_2^2$ and the upper bound for $\|w_t^1\|_2^2$ and $\|w_t^2\|_2^2$. Then we conclude that

$$\|X_t\|_2^2 - \|X_{t+1}\|_2^2 \geq (\|v_t^1\|_2^2 + \|v_t^2\|_2^2) - (\|v_t^1\|_2^2 + \lambda^2\|v_t^2\|_2^2 + \frac{5}{8}\delta(1-\lambda^2))$$

$$\geq \delta(1-\lambda^2) - \frac{5}{8}\delta(1-\lambda^2)$$

$$= \frac{3}{8}\delta(1-\lambda^2) > \frac{1}{8}\delta(1-\lambda^2) = \epsilon,$$

where a contradiction appears.

This completes the proof. $\qquad\square$

# G  More Experiments

We provide additional experiments to demonstrate the behaviors of the optimistic gradient and negative momentum methods in convergent perturbed games that do not satisfy the BAP assumption. The numerical results reveal cases where optimistic gradient/momentum method converge and cases where they do not converge.

In the same setting as the experiments on Theorem 3.4, we find that both optimistic gradient descent ascent and negative momentum method converge as shown in Figure (8).

However, there are other cases in which these two algorithms do not converge. In Figure (9), we present one such example. Here the payoff matrix is chosen as $A = [[1,0],[0,0]], B = [[0,8],[0,0]]$ and

$$A_t = \begin{cases} A, & t \text{ is odd} \\ A + (1/t^{0.1}) * B, & t \text{ is even} \end{cases}. \tag{35}$$

In Figure (9), the numerical results show when using a step size of $0.015$, optimistic gradient and negative momentum algorithms will diverge, but extra gradient will converge. Based on these numerical results, we believe that beyond the setting that satisfies the BAP assumption, there exists a more complex dynamical behaviors of optimistic gradient and negative momentum methods, which presents an interesting question for future exploration.

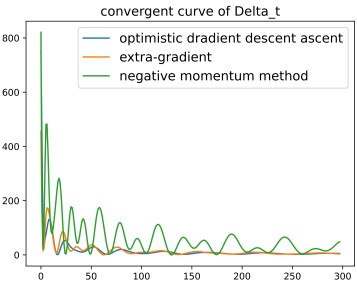

**Figure 8:** Function curves of $\Delta_t$ for one game presented in experiment of Theorem 3.4. in the paper. All these three algorithms converge.

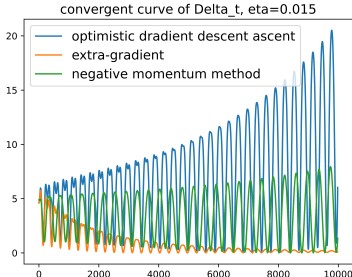

**Figure 9:** Function curves of $\Delta_t$. When using step size = 0.015, extra-gradient converges, while both optimistic gradient descent ascent and negative momentum method diverge.