# OpenReview forum: "On the Last-iterate Convergence in Time-varying Zero-sum Games: Extra Gradient Succeeds where Optimism Fails"
_NeurIPS.cc/2023/Conference — NeurIPS 2023 poster_

### Official Review · Reviewer_tqyN · 2023-06-18

**Soundness:** 3 good
**Presentation:** 3 good
**Contribution:** 3 good
**Rating:** 5
**Confidence:** 3

**Summary:**

This paper considers the problem of unconstrained min-max optimization with bilinear structure. Specifically, this paper considers the setting where the payoff matrix $A_t$ changes over time. The two changing dynamics of $A_t$ considered in this paper is the periodic game and the converging perturbed game. Three strategy dynamics are cosidered in this paper, which is OGDA, extra gradient (EG) and Negative momentum method. In this paper, the authors show that in the period game case, under certain initialization, OGDA provably diverges no matter what learning rate is chosen while EG with a certain choice of learning rate converges to the common NE with exponential rate. In the convergent perturbed game case, all the three types of algorithms converge with a proper choice of learning rate. The analysis is basd on analyzing the eigenvalues of the linear operator in the recursive form of the strategies. Empirical results further verifies their theoretical results.

**Strengths:**

- The min-max optimization with time-varying functions is an important question and has wide real-world applications.
- This paper first considers the last-iterate convergence performance for the unconstrained bilinear min-max problem and show that under two different changing dynamics of the bilinear matrix, OGDA performs provably differently and EG always converges. Specifically, the diverging results for OGDA looks intereting to me.
- I did not check all the proof details but the general idea of the proof makes sense to me.

**Weaknesses:**

- One issue is whether the optimization of unconstrained min-max optimization with bilinear structure is interesting. Specifically, even when the game is changing over time, one trivial Nash Equilibrium is $(0,0)$. Therefore, all the three algorithms initialized with $(0,0)$ will converge, though the dynamic of these three algorithms may still be interesting with different initializations. Extending the current framework to a slightly general case $x^\top A_ty+b_t^\top x+c_t^\top y$ or an even more general convex-concave function $f(x,y)$ will strengthen the results a lot.
- As the authors mentioned, the constrained version of the min-max optimization problem may be more interesting and the dynamic of the strategy can not be expressed in a clean linear recurrsive form as shown in the unconstrained bilinear case.

**Questions:**

- As mentioned in weakness, whether the current result can be extended to a generalized linear case with $f_t(x,y)=x^\top A_ty+b_t^\top x+c_t^\top y$?
- Is there a specific reason considering the periodic games and the convergent perturbed games? Can the results generalize to some other type of time-varying games with common NE?

**Limitations:**

See weakness and questions for details.

---

> ### Author Rebuttal · Authors · 2023-08-10
>
> Thanks a lot for your support and interests to our results, especially we thank you for proposing important and challenging question to strengthen current results. Please see our itemized responses below:
>
> 1. Whether the current result can be extended to a generalized linear case with $f_t(x,y) = x^{\top}A_ty + b^{\top}_tx + c^{\top}_ty $ ?
>
> Thank you for asking this important question. Generalizing the current results to other settings, such as convex-concave payoff, is definitely an important question for future research. Our results can be extended to the above general linear case, as the linear payoff can be reduced to bilinear case with translation of the strategy variables. Similar technologies were also introduced in (Zhang et al. convergence of gradient methods on bilinear zero-sum games) in the stationary game setting. In the following we provide a more detailed explanation.  For convenience we formulate the payoff function as
>  \begin{align}%\label{gene}
> f_t(x,y) = x^{\top}A\_ty + x^{\top}b\_t + c^{\top}\_ty. \\ \\ \\ (1)
> \end{align}
> By first order stationary condition, if a point $(x^*,y^*)$ is a equilibrium of $f_t(x,y)$, then
> \begin{align}%\label{equi}
>         A\_ty^* = - b\_t, \  A^{\top}\_t x^* = -c\_t.\\ \\ \\ (2)
> \end{align}
>
> For a periodic game with payoff matrix $\\{A\_t\\}^{\infty}\_{t=1}$, payoff vectors $\\{b\_t\\}^{\infty}\_{t=1}, \\{c\_t\\}^{\infty}\_{t=1}$ and period $\mathcal{T}$, if a strategy point $(x^*,y^*)$ is a common equilibrium, then it satisfies (2) for any $t \in [\mathcal{T}]$. We assume such common equilibrium exists, thus there exist $(b,c)$ such that
>     \begin{align}%\label{111}
>         A\_t b = - b\_t, \  A^{\top}\_t c = -c\_t,\  \forall t \in [\mathcal{T}]. \\ \\ \\ (3)
>     \end{align}
>     Now we denote $x' = x - c,\ y' = y - b$, then it can be verified that
>     \begin{align}
>         \min\_x \max\_y f\_t(x,y) = \min\_{x'} \max_{y'} (x')^{\top}A\_ty'+c^{\top}A\_tb.
>     \end{align}
> Thus every periodic game with general linear payoff is an equivalent to a bilinear periodic game considered in the current paper, thus the result in Theorem 3.1. also holds in general linear payoff case. However, according to (3), it may be difficult for a periodic game with a general linear payoff to have a common equilibrium. This is different from the bilinear game where $(0,0)$ is always a common equilibrium.
>
> For convergent perturbed game, it can be demonstrated that Theorem 3.3. still holds true with a similar translation of the variables and techniques employed in the paper.
>
> ****
> 2. Is there a specific reason considering the periodic games and the convergent perturbed games? Can the results generalize to some other type of time-varying games with common NE?
>
> For the first question, please refer to the second part of the global rebuttal.
>
> For the second question, generalizing the current results to other type of time-varying games with common NE is a very interesting question. It is not clear to us if our results would carry over for more general class of games with common NE and if our techniques will be applicable. An assumption that probably is needed if one wants to show convergence (e.g., of extra gradient), is that the difference $||A_{t+1}-A_t||_2$ is not too large for all $t$. We would like to mention that general theory of non-autonomous linear differential/difference equation is not as simple as autonomous ones, it is still a quite open-ended area by itself (see Fritz Colonius and Wolfgang Kliemann. Dynamical systems and linear algebra, volume 158.344 American Mathematical Society, 2014.). Thus we believe that any interesting generalization will depends on some specific observations case by case, which means there are many potential problems to be answered.

---

### Official Review · Reviewer_PugT · 2023-07-06

**Soundness:** 3 good
**Presentation:** 3 good
**Contribution:** 3 good
**Rating:** 6
**Confidence:** 3

**Summary:**

This paper studies the last-iterate behavior of three different algorithms on two types of time-varying zero-sum games: periodic and convergent perturbed games. For periodic games, the authors prove that EG will converge while OGDA and the negative momentum method could diverge. For convergent perturbed games, all these algorithms converge as long as the error term decays sufficiently fast.

**Strengths:**

1. The paper is well-organized and easy to follow.
2. The different behavior between EG and OGDA seems novel in the literature.

**Weaknesses:**

1. Some results lack an intuitive explanation.
2. Some figures are not well-plotted. Maybe a log-log graph is more proper.

**Questions:**

1. Could the authors provide some examples of periodic and convergent perturbed games?

2. In line 225, the authors say that the key point on the convergence of EG for periodic games is that the iterative matrix is a normal matrix. Could the authors give a more intuitive explanation? Moreover, the update rule of EG only depends on the current point $(x_t, y_t)$, while OGDA and negative momentum method also need to employ the gradient information at the last point $(x_{t-1}, y_{t-1})$. Is this a reason for the different behavior of these methods?

3. For the experiments on Theorem 3.4, the authors only plot the convergent curve of EG. I wonder about the behabior of the other two methods in this case.


**Limitations:**

See Questions.

---

> ### Author Rebuttal · Authors · 2023-08-10
>
> Thank you for your supportive comments, suggestions and questions on intuitive explanation and experiments. Please see our itemized responses below:
>
> 1. Could the authors provide some examples of periodic and convergent perturbed games?
>
> Please refer to part 2 of the global rebuttal for an explanations of this question.
> ****
>
> 2. Intuitive explanation for why EG's iterative matrix is a normal matrix.
>
> Thank you for asking this important question. "EG's iterative matrix is normal" is an observation that is also surprising to authors. Our intuition actually came from experiments. First of all, we expect that all iterative matrices to be diagonalizable. However, we observe that the dynamics never diverges, which is a much stronger phenomenon than being diagonalizable. Since normal matrices has eigenvalues with magnitudes at most 1, and this means their dynamics never diverge, which agreed with all the experiments we had run. Therefore, we turn our attention from diagonalizable to normal and eventually discover only EG is normal, and it became the key property for periodic settings.
>
> For the differences between EG and optimistic/momentum methods, other than their disparities in eigenvalues/eigenvectors. The suggestion that the differences in dependence between the update rules and past payoffs is an interesting viewpoint, we will consider this to see whether it can give an intuitive explanation. We also verify that the iterative matrix of gradient descent algorithm, which also uses only one step information is normal matrix.
> ****
> 3. Behaviors of the other two methods in this case in the experiments on Theorem 3.4,
>
> The other two methods also converge under the conditions of experiments on Theorem 3.4. However, we have also found a numerical example that does not satisfy the BAP assumption, and it seems that the optimistic gradient/momentum method does not converge in this example. Please refer to the first part of global rebuttal for additional experiments and explanations of this question.

---

> > ### Comment · Reviewer_PugT · 2023-08-12
> > **Thanks for the replies.**
> >
> > Thanks for the detailed replies, which have addressed my concerns. I increase my score to 6.

---

> > > ### Author Response · Authors · 2023-08-12
> > >
> > > Thank you for your interest to the work and intriguing questions. We appreciate your boosting to the score, thank you so much!

---

### Official Review · Reviewer_qpgT · 2023-07-07

**Soundness:** 3 good
**Presentation:** 3 good
**Contribution:** 3 good
**Rating:** 7
**Confidence:** 3

**Summary:**

This paper studies the problem of learning Nash equilibria in two-player zero-sum bilinear games where the payoff matrix varies with time.
When the payoff matrix is a periodic function, it is proven that the extra gradient algorithm converges to a Nash equilibrium, whereas the optimistic gradient descent ascent and negative momentum method could diverge from the equilibrium.
Furthermore, it is shown that all the above three algorithms converge to the equilibrium for the convergent perturbed game.

**Strengths:**

* The paper is well-written and studies an interesting problem.
* The author provided an instance of periodic zero-sum games where extra gradient algorithms and optimistic gradient descent ascent algorithms behave distinctly differently.
* The analysis looks to be sound, and the experimental results seem to bare out the theory.

**Weaknesses:**

* Perhaps a short intuitive explanation of why extra gradient is better than optimistic gradient descent ascent would help readers understand the main results of this paper.
* The titles and legends in Figures 1,2,3 and 4 should be enlarged.

**Questions:**

* (Section 2.1) What is the definition of $\mathrm{ker}(\cdot)$?
* (Section 3.2) What are the key differences between the extra gradient algorithm and the other two algorithms?
* (Section 3.2) For the periodic games with $\lambda_{\ast}=0$, would similar convergence/divergence results hold to Theorem 3.1 and 3.2?
* (Section 4) Do the optimistic gradient descent ascent algorithm and negative momentum method empirically converge in the same instance in the experiments for Figure 4?

**Limitations:**

The authors adequately addressed the limitations.

---

> ### Author Rebuttal · Authors · 2023-08-10
>
> Thank you very much for your support, careful reading and helpful suggestions. Please see our itemized responses below:
>
> 1. What is the definition of $\ker(\cdot)$ ？
>
> For a payoff matrix $A \in \mathbb{R}^{n \times m}$, $\ker(A)$ is defined to be the set $\\{ x \in \mathbb{R}^m |\ Ax = 0  \\}.$
>
> ****
>
> 2. What are the key differences between the extra gradient algorithm and the other two algorithms?
>
> *A quick answer: Only the iterative matrix of extra gradient is a normal matrix, while the other algorithms' iterative matrices are just diagonalizable but not normal.
>
> *A detailed answer: From the techniques we used to prove the separation between extra gradient and the other two methods, the key difference is that the iterative matrix of extra gradient algorithm is a normal matrix,  while the other two algorithms are not.  For example, in the period case, the advantages of normality are twofold : it helps to bound the modulus of eigenvalues of iterative matrix and makes the strategy components that do not correspond to equilibrium vanish over time. In the following, we provide further explanations.
>
> Firstly, let's denote the overall iterative matrix as $\prod_{t=1}^T \mathcal{A}\_t$, which is product of iterative matrix for a period of $T$. For matrix $\prod_{t=1}^T \mathcal{A}_t$, the maximum modulus of eigenvalues is no more than 1 due to the normality, thus during the iterative process, the strategy can't diverge.
>
> Secondly,  we introduce a decomposition of the two players' current strategy $X_t$ into two parts. The first component $X_t^1$ represents eigenvector corresponding to eigenvalue $1$ of the iterative matrix, which corresponds to an equilibrium of the game. The second component $X_t^2$ consists of linear combinations of other eigenvectors. Due to the normality property of the iterative matrix of extra gradient, $X_t^1$ remains unchanged over time and modulus of $X_t^2$ approaches zero as time goes to infinity. (See Corollary B.4 and proof of Theorem 1 in Appendix B).  This ensures convergence of extra gradient algorithm towards Nash Equilibrium point. For the other two algorithms, without the normality property, for the overall iterative matrix, as we shown in the example (See Theorem 3.2 and its proof in Appendix C), its maximum modulus of eigenvalues can be larger than 1. Then, we can choose the initial point which makes the strategy point diverge from Nash Equilibrium point. The normality of iterative matrix of extra gradient also plays important role in convergent perturbed game.
>
> ****
> 3. For the periodic games with $\lambda_* = 0$, would similar convergence or divergence results hold to Theorem 3.1 and 3.2?
>
> Yes, the result still hold, since in this case from the general result in linear algebra, the iterate matrix $\prod^T_{t=1} \mathcal{A}_t$ will be a nilpotent matrix, which means after finite time iteration, the strategy $(x_t,y_t)$ must come to be the trivial equilibrium $(0,0)$.
>
> The divergence result in Theorem 3.2 depend on the the fact that for the special periodic game defined by the payoff matrix
> \begin{align}
>  A_t=
> \begin{cases}
> \left[1,-1\right], & t \textnormal{\ \ is \ odd} \\\\
> \left[-1,1\right], & t\textnormal{\ \ is \ even}
> \end{cases},
> \end{align}
> the iterate matrix of optimistic gradient and negative momentum method always have a eigenvalue with modulus larger than $1$, combine with Floquet theorem (proposition 2.4, line 166), this is enough to show the exponential divergence result, and whether $\lambda_* = 0$ is irrelevant for Theorem 3.2.
> ****
> 4. Do the optimistic gradient descent ascent algorithm and negative momentum method empirically converge in the same instance in the experiments for Figure 4?
>
> Yes, both the optimistic gradient descent ascent algorithm and the negative momentum method also converge in the same instance in the experiments for Figure 4. However, we also find examples where these two methods do not empirically converge due to violations of the BAP assumption, please refer to part 1 in the global rebuttal for detailed explanation.
> ****
> Thank you for your comments on the titles and legends in the figures, we will address these issues in the new version of the paper.

---

### Official Review · Reviewer_oHHw · 2023-07-09

**Soundness:** 3 good
**Presentation:** 3 good
**Contribution:** 3 good
**Rating:** 7
**Confidence:** 3

**Summary:**

Over the last few years an extensive literature has studied the last iterate convergence of learning dynamics in zero-sum games, particularly those using optimism and extra-gradient approaches.  This paper extends that approach to two classes of unconstrained non-stationary game (periodic and decaying noise) and shows a difference in behavior between the two classes.

**Strengths:**

The problem studied is natural and clearly explained with strong results, including the interesting split in behavior (Theorems 3.1 and 3.2)

The approach uses several technical innovations derived from the literature on non-autonomous linear difference systems.


**Weaknesses:**

The particular classes of games studied could be better motivated.  Is there a reason to focus on these other than technical convenience?  Do they have important applications?  What more general but harder class is this a step towards?  The conclusion mentions constrained games as future work, nothing about other interesting directions fof future work.  Are there other theorems from non-autonomous linear difference systems that seem promising?

There is a gap left by Theorems 3.3 and 3.4 about whether the other two dynamics converge under the weaker conditions of Theorem 3.4.  Figure 4 gives an experiment on this setting, so it seems like a missed opportunity to not at least show what the other dynamics do on this example.

Minor comments:
- There is some inconsistency in the way “equilibrium” is used.  It is defined on line 98 as what I would instead have called the set of equilibria.  However, in other places, like line 252 it seems to be used to refer to a single element of this set.  Depending on how this is resolved, some care may be needed in places about whether something is “the” or “a” equilibrium (e.g. on line 216).
- Line 178 and 181 the formulas use n but the limits are taken over t
- Line 216 the “common Nash equilibrium” is not defined, nor is there an explanation of why it mush exist (presumably because 0 is always an equilibrium?)
- Line 305 group[s]


**Questions:**

Please respond regarding the two main weaknesses.

**Limitations:**

The conditions to which the results apply are clearly explained.

---

> ### Author Rebuttal · Authors · 2023-08-10
>
> Thank you for your support, interest to current work, and inspiring question for future work. Please see our itemized responses below:
>
> 1. Is there a reason to focus on these classes of games other than technical convenience? Do they have important applications? What more general but harder class is this a step towards?
>
> For the first and second question, please refer to the second part in global rebuttal for a detailed explanation.
>
> For the third question, we believe that the class of convergent perturbed games with a convergence rate slower than $\frac{1}{t}$ is more challenging compared to our current setting. As discussed in Section 5, the dynamical behaviors of optimistic gradient and negative momentum methods under this setting remain unclear. Based on the numerical results presented in the experiments in global rebuttal, we believe that the dynamical behaviors of optimistic gradient and negative momentum method under this setting are much more complex than those in the BAP setting, as stated in Theorem 3.3.
>
> ****
> 2. The conclusion mentions constrained games as future work, nothing about other interesting directions of future work. Are there other theorems from non-autonomous linear difference systems that seem promising?
>
> Stability theory of non-autonomous dynamical system from learning algorithms has been studied recent years, e.g. "AdaGrad Avoid saddle points, Antonakopoulos et al." and "First-order method almost always avoid saddle points:the case of vanishing step-size, Panageas et al.". Both works investigate non-autonomous dynamical systems from non-convex minimization problems. Their method belongs to a general theoretical framework called Lypunov-Perron method and stable manifold theorem, which enables one to prove certain dynamical systems do not converge to unstable fixed points. One promising future direction is to fit such method into min-max optimization algorithms so that we can have a better understanding on the asymptotic behavior of different algorithms in time-varying settings. One might expect some result similar to "The limit points of (optimistic) gradient descent in Min-Max optimization, Daskalakis et al.", but in a time-varying setting. Since the Lyapunov-Perron method depends on spectral analysis of update rules (spectral analysis is a main technique in all  aforementioned papers), it is interesting to explore if these techniques can be combined with current paper.
>  ****
> 3. Gap between Theorem 3.3 and Theorem 3.4.
>
> Please refer to the first part of the global rebuttal for additional numerical results and explanations regarding this question.
>
> ****
> Thank you for the other comments on writing, the typos will be fixed in the new version of the paper.

---

> > ### Comment · Reviewer_oHHw · 2023-08-11
> > **Thanks**
> >
> > This addresses my questions

---

> > > ### Author Response · Authors · 2023-08-12
> > >
> > > We are glad to answer your questions, thank you for proposing them. Thanks again for your strong support!

---

### Official Review · Reviewer_abUR · 2023-07-22

**Soundness:** 3 good
**Presentation:** 3 good
**Contribution:** 2 fair
**Rating:** 5
**Confidence:** 3

**Summary:**

This work investigates the problem of last-iterate convergence in time-varying zero-sum games. Specifically, the authors study the last-iterate convergence of three kinds of algorithms (OGDA, EG, and negative momentum method) considering two kinds of time-varying games with specific structures, i.e., periodic games and convergent perturbed games. The authors obtained a convergence rate of EG in periodic games while providing a counterexample showing that OGDA and negative momentum method will diverge in this kind of game. In convergent perturbed games, the authors showed that all three kinds of algorithm will converge with a rate dependent on the perturbation $B_t$. Finally, experiments validate the statements proposed by the authors.

**Strengths:**

In general, I think there is no last-iterate convergence rate in time-varying games due to the changing nature of the games. However, with specific problem structures, I believe that last-iterate convergence results can be established, and this work provides a piece of clear evidence. Overall, although the solutions are relatively simple by modeling the learning algorithms as dynamical systems and leveraging some existing results in the control theory, I am satisfied with the motivation and the final results of this work.

**Weaknesses:**

There are no significant weaknesses that affect my rating of the whole paper. In the following, I only list some minor issues.

1. The notion of $T$ in periodic games. The notion of $T$ is commonly used to refer to the total number of rounds in online learning. In this work, the authors use $T$ to represent the number of rounds inside each period, which may lead to unnecessary understandings. I suggest the authors could choose a different notion to denote the period length.
2. The notion of $\text{ker}(\cdot)$. This notion, which appears in Line 98 for the first time, seems not to be defined or given a clear description before.
3. In Line 120, the authors mention that 'Recently, there are also works analyzing the regret behaviors of OGDA under a time-varying setting [1]'. The reference to [1] is not accurate enough. The work of [33], which is published at ICML2022, firstly gives a comprehensive study of the optimistic methods in time-varying zero-sum games by optimizing multiple performance measures simultaneously. I suggest the authors could refer to [33] in the aforementioned statement to make the credits more accurate.
4. The negative momentum method requires the $x$-player to evolve to the next round first to give its gradient $A_{t+1} x_{t+1}$ to the $y$-player. In some cases where the learning procedure is strictly in an online style, this algorithm is NOT applicable since both players must act first to make the game evolve. I suggest the authors could give a discussion on this point in the revised version.

Typos:
1. Line 89: 'Section 4' not 'section 4'.
2. Line 188: 'Section 3.3' not 'Section3.3'.
3. The reference to theorems are not unified, e.g., in Line 237, 'theorem 3.2', and in Line 255, 'Theorem (3.1)'.

**Questions:**

I only have a major question on Theorem 3.1. What is the range of $t$? If $t$ is only chosen in $[1,T]$, choosing $t=T$ yields a convergence rate of $O(\lambda_* \text{poly}(T))$, which can be seen as a constant in terms of $T$. As mentioned in the 'Weaknesses' part, a bound linear in $T$ is pretty bad in the standard online learning convention. Theorem 3.1 exhibits an exponential rate only when $t$ can be significantly larger than the period length. Can the authors give some further explanations on this point?

In the end, I am very curious about whether the method in this paper, which models the learning algorithms as dynamical systems, can be applied to the constrained case, where the projection operations may bring some unique challenges to the modeling issue. Can the authors give some further explanations on this point?

**Limitations:**

Please mainly refer to the 'Weaknesses' part.

---

> ### Author Rebuttal · Authors · 2023-08-10
>
> Thank you for your careful reading, supportive comments, and helpful suggestions in improving the paper. We will clarify the issues in revision. Please see our itemized responses below:
>
> Questions part：
>
> 1. What is the range of $t$ in Theorem 3.1 ?
>
> Here $t$ denotes the number of rounds that players have played in the game, thus $t$ takes value in range $[0,+\infty)$ and $T$ denote the length of the period, which can be thought as a constant for a given periodic game. In Theorem 3.1 we provide a $\mathcal{O}( (\lambda_*)^{t/T} \cdot \text{Poly}(t) )$ with $ \lambda_* <1$ bound on the distance between the current strategy and equilibrium. Therefore, when two players use extra-gradient, both players' current strategies will converge to equilibrium with an exponential rate.
> ****
> 2. Whether the method in this paper, which models the learning algorithms as dynamical systems, can be applied to the constrained case ?
>
> Learning algorithms in which there are constraints (like simplex constraints in games) can be analyzed using techniques from dynamical systems (e.g., see Last-Iterate Convergence: Zero-Sum Games and Constrained Min-Max Optimization, Daskalakis et al). As long as the NE is in the interior of the constrained set, the same techniques as used in this paper are applicable. The main challenge occurs when the NE is on the boundary of the constrained set. Our techniques can be applied in the constrained case as long as the constrained set can be expressed as a polytope (linear equalities and inequalities). In the more general case in which the set is an arbitrary convex set, more involved techniques are needed and further assumptions (assumptions on the curvature of the constrained set, see for example analysis of manifold gradient descent in First-order Methods Almost Always Avoid Saddle Points Lee et al.). This is an interesting open question for future investigation, though we believe similar results will occur. There are several techniques from differential geometry that deal with constraints (see for example analysis of manifold gradient descent in First-order Methods Almost Always Avoid Saddle Points Lee et al. which expresses manifold gradient descent as a dynamical system). As long as the constrained set is a smooth compact submanifold, one can use the projection operator and Riemannian gradient to express the learning dynamics as a dynamical system. Our techniques can deal with polytope constraints, it is very interesting future question to generalize these results to more general constrained convex sets.
> ****
> Weaknesses part：
>
> Thank you for your helpful comments.
>
> 4. Notion of $T$
>
> In the new version of the paper we will use $\mathcal {T}$ to represent the length of period in a periodic game.
> ****
> 5. Definition of $\ker(\cdot)$
>
> We will add a formal definition of $\ker(\cdot)$ in line 98 in the paper. Here $\ker(A) = \\{ x \in \mathbb{R}^m | Ax = 0  \\}$.
> ****
> 6. Inaccurate reference
>
> Thank you for your pointing out this issue. We apologize for our inattention. In the new version we will refer [33] in line 120 to ensure more accurate credits.
> ****
> 7. Question about $A_{t+1}x_{t+1}$ in negative momentum method.
>
> Here we are using the $\textbf{alternating}$ negative momentum method studied in (Gidel et al. Negative Momentum for Improved Game Dynamics) instead of the usual $\textbf{simultaneous}$ setting. In the alternating setting, at  round $t$, the $y$-player first chooses his strategy $y_t$ based on the payoff caused by $A^{\top}\_{t-1}x\_{t-1}$, and then the $x$-player choose her strategy $x_t$ based on the payoff caused by $A_{t}y_{t}$. In the stationary game setting, (Gidel et al. ) proved that the simultaneous negative momentum method will lead to an exponential divergence rate, while the alternating negative momentum method will converge. Since our focus in this paper is on determining which algorithm will converge in a time-varying game, we only consider the alternating setting.
>
> ****
> Typos will be corrected in the new version of the paper, thank you for pointing them out.

---

> > ### Comment · Reviewer_abUR · 2023-08-11
> > **Thanks for the explanations**
> >
> > Thanks for the detailed feedback, which has fully answered my questions.

---

> > > ### Author Response · Authors · 2023-08-12
> > >
> > > You are very welcome! Thanks a lot for reading and agreeing with the response.

---

### Author Rebuttal · Authors · 2023-08-10

We appreciate the efforts of all reviewers, thanks for your constructive questions and critical suggestions! The PDF file contains additional experimental results as requested by reviewers. If you have questions about the experimental parts of the paper, please refer to this file.  In the following, we address two questions that have been raised by several reviewers.
****
1. Gap between Theorem 3.3 and Theorem 3.4., and behaviors of optimistic gradient/momentum method in the instance in Figure 4 of the paper.

We provide additional experiments to demonstrate the behaviors of the optimistic gradient and negative momentum methods in convergent perturbed games that do not satisfy the BAP assumption. The numerical results reveal cases where optimistic gradient/momentum method converge ( Figure 1 in PDF records experimental results) and cases where they do not converge (Figure 2,3 in PDF records experimental results).

In the same setting as the experiments on Theorem 3.4 , we find that both optimistic gradient descent ascent and negative momentum method converge as shown in Figure 1 of the PDF records experimental results.

However, there are other cases in which these two algorithms do not converge. In Figure 2 and 3, we present one such example. Here the payoff matrix is chosen as $A = [ [1,0],[0,0]], B = [[0,8],[0,0]]$ and

\begin{align}
 A_t=
\begin{cases}
A, & t \textnormal{\ \ is \ odd} \\\\
A + (1/t^{0.1}) * B , & t\textnormal{\ \ is \ even}
\end{cases}.
\end{align}

 In Figure 2, the numerical results show when using a step size of $0.015$, optimistic gradient and negative momentum algorithms will diverge, but extra gradient will converge. In Figure 3, by reducing the step size to very small numbers ($0.0003$ or $0.0001$ in this case),  optimistic gradient and negative momentum algorithms do not seem to diverge but maintain a nonzero distance from the equilibrium points. Based on these numerical results, we believe that beyond the setting that satisfies the BAP assumption mentioned in line 256 of the paper, there exists a more complex relationship between the dynamical behaviors of optimistic gradient and negative momentum methods and their respective step sizes, which presents an interesting question for future exploration.

****
2. Reasons to consider periodic/convergent perturbed game and examples of them.

We note that both of these game classes have been studied in previous literature as testing grounds of learning algorithms in online learning community ([11], [14]). However, our paper focuses on different viewpoints compared to the existing research, making it a natural extension of this line of research. Moreover, periodic game and convergent perturbed game are natural generalizations of the usual repeated game formulations. When the period $\mathcal{T} = 1$, the periodic game becomes a repeated game, and when the perturbation $B_t = 0$, then convergent perturbed game becomes a repeated game.

The periodic games model competitive settings where the exogenous environment varies periodically. This naturally occurs in competitions where day-to-day trends, and seasonality can affect the game between players. For example, consider a competition between two species whose life states are affected by seasonal changes or sellers in a fish market, where the value of fish depends on their freshness, thus exhibiting daily period behavior. Periodic zero-sum games can also fit into the frameworks of multi-agent contextual games (Sessa et al. Contextual games: Multi-agent learning with side information, NIPS 2020). In a multi-agent contextual game, the environment selects a context from a set before each round of play and this selection determines the specific game that will be played. Periodic zero-sum games can be viewed as a multi-agent contextual game where the environment periodically chooses contexts from the available set, creating zero-sum games with a common equilibrium. The convergent perturbed game naturally models games with noise that decays over time. The noise can arise from players' own beliefs, such as in an auction where a certain type of goods is being auctioned off day after day to  buyers, and the buyers' assessment of the value of this good will eventually converge to a fixed value as their experience growth. The noise can also model external effects when players repeatedly play the game, such as interference factors in the feedback process.

****
We are looking forward to further discussion in the next stage of review process. We thank your again for all your hard work!

---

### Decision · Program_Chairs · 2023-09-21

**Decision:**

Accept (poster)

**Comment:**

This paper shows a very interesting and somewhat surprising result: that the extragradient method may have stronger convergence properties in certain zero-sum games than optimistic gradient descent-ascent. This is surprising because most results are essentially the same for both algorithms, and people even sometimes refer to OGDA as a variant of "single-call extragradient."

At the same time, a reviewer pointed out a somewhat significant weakness: that bilinear games are somewhat trivial, in that (0,0) is always an equilibrium. Despite the weakness, all the reviewers are in favor of acceptance. Drawing an analogy to the trajectory of work on last-iterate convergence in zero-sum games, one might also expect that follow-up work will be able to move to more realistic and less trivial games.